# Photo-activated raster scanning thermal imaging at sub-diffraction resolution

M. Bouzin[1,6], M. Marini[1,6], A. Zeynali [1], M. Borzenkov[2], L. Sironi[1], L. D'Alfonso[1], F. Mingozzi[3], F. Granucci[3], P. Pallavicini[4], G. Chirico [1,5]* & M. Collini[1,5]

Active thermal imaging is a valuable tool for the nondestructive characterization of the morphological properties and the functional state of biological tissues and synthetic materials. However, state-of-the-art techniques do not typically combine the required high spatial resolution over extended fields of view with the quantification of temperature variations. Here, we demonstrate quantitative far-infrared photo-thermal imaging at sub-diffraction resolution over millimeter-sized fields of view. Our approach combines the sample absorption of modulated raster-scanned laser light with the automated localization of the laser-induced temperature variations imaged by a thermal camera. With temperature increments ∼0.5–5 °C, we achieve a six-time gain with respect to our 350-μm diffraction-limited resolution with proof-of-principle experiments on synthetic samples. We finally demonstrate the biological relevance of sub-diffraction thermal imaging by retrieving temperature-based super-resolution maps of the distribution of Prussian blue nanocubes across explanted murine skin biopsies.

[1] Physics Department, Università degli Studi di Milano-Bicocca, Piazza della Scienza 3, 20126 Milano, Italy. [2] Medicine and Surgery Department, Nanomedicine Center, Università degli Studi di Milano-Bicocca, Piazza della Scienza 3, 20126 Milano, Italy. [3] Biotechnology and Biosciences Department, Università degli Studi di Milano-Bicocca, Piazza della Scienza 2, 20126 Milano, Italy. [4] Chemistry Department, Università degli Studi di Pavia, Viale Taramelli 12, 27100 Pavia, Italy. [5] CNR Institute for Applied Science and Intelligent Systems, Via Campi Flegrei 34, 80078 Pozzuoli, Italy. [6]These authors contributed equally: M. Bouzin, M. Marini. *email: giuseppe.chirico@unimib.it

Active photo-thermal imaging over submillimeter-sized to centimeter-sized fields of view with tunable spatial resolution in the 10–100 μm range would be a valuable tool for the nondestructive characterization of the morphology and functional state of both biological tissues and synthetic materials. Differently from passive techniques, which measure temperature changes spontaneously occurring in the sample due to intrinsic heat generation, active approaches rely on monitoring the peculiar sample thermal response upon application of an external heating (or cooling) stimulus[1–4]. In the clinical and pre-clinical setting, for example, active medical thermography allows identifying the modified blood perfusion and metabolic activity associated to the presence of inflammations, cancer masses or physiological dysfunctions by accessing the resulting alterations in the tissues thermo-physical properties (e.g., the thermal conductivity[3], or the sample temperature during the thermal relaxation phase[2]). Externally induced temperature variations also provide direct access to the spatial distribution of bio-markers, as demonstrated with melanin for the identification of age-related macular degeneration in the retina[5] or the discrimination of in-situ and invasive malignant skin lesions in the context of melanoma screening and diagnosis[6,7]. In nanomedicine and nanotechnological research, photo-induced heat release events allow characterizing the distribution of nanoparticles following systemic administration in animal model systems, and the development of related photo-thermal therapy protocols[8,9].

Overall, active medical thermography finds promising applications ranging from the development of quantitative screening tools for the (early-stage) detection of pathologies, to the characterization of disease progression and therapy response in forefront pre-clinical research. High resolution active thermal imaging is equally relevant to monitor the dissipation efficiency or detect the fault of electronic devices and micro-electro-mechanical systems[10,11], or to perform quality controls on the thermo-conduction properties of materials[4,12]. For all these applications, morphological imaging of the sample structure with resolution well below the millimeter range should be combined with the high sensitivity (∼0.1 °C) measurement of the induced temperature variations over millimeter to centimeter-sized fields of view.

Existing approaches reaching the highest (sub-micrometer) spatial resolution include fluorescence-based thermometry, where temperature is probed via its effect on the intensity, anisotropy and lifetime of the fluorescence emission of dyes, proteins and nano-constructs[13–16], and scanning thermal microscopy, that relies on the near-field interaction between a surface and a heated tip[17–21]. While being applicable in principle to the detection of externally (e.g., optically or electrically) induced temperature variations, these techniques are usually exploited to map the intrinsic temperature in the sample with sub-micrometer resolution over relatively small (<100 × 100 μm²) regions. On the other hand, in the context of active thermography, the exploitation of the response of optical properties (reflectivity and index of refraction) to a modulated change in the sample temperature is at the basis of scanning thermo-reflectance microscopy[11,22–24] and PHoto-thermal Imaging (PHI)[7,25–28]. In the case of PHI, temperature increments are indirectly monitored by the variations in the refractive index: as a result, PHI images allow mapping the absorbers distribution, but cannot easily be converted into temperature maps. Scanning thermo-reflectance imaging can reach nanoscale spatial resolution and provide quantitative measurement of the induced temperature variations after sample-dependent calibration[11]. It finds however its best application in the characterization of electronic and opto-electronic devices[11,29]; the sample coating with a thin metal transducer layer, which is often required to

improve the technique sensitivity[30–32], may limit the applicability to biological systems.

More straightforward access to temperature values, coupled to imaging over wider areas, can be achieved by infrared thermal imaging[33–35]. Upon calibration of the sample emissivity[33,34,36], temperature increments can be quantified based on Stefan-Boltzmann's law by the measurement of the infrared radiance in the gray-body approximation. However, the typical ∼1–100 °C range of interest corresponds to peak radiances in the ∼7–11 μm band[34]: even though expensive thermo-cameras with pixel size as low as 5–20 μm on the sample plane exist in commerce, the effective resolution is limited to ∼0.1–1 mm[37–40] due to the thermal waves diffusion in the sample and to the diffraction of the far-infrared radiation at the low numerical aperture of Germanium or Zinc-Selenide optics (Abbe's law[41–43]). Higher spatial resolution (∼100 μm) and <0.1 °C sensitivity have been reached in specially designed military equipment only[44,45].

It is therefore our purpose here to develop and validate a non-contact super-resolution infrared thermal imaging approach capable of quantitatively measuring temperature increments, as allowed by conventional thermography, and of simultaneously achieving the ∼50 μm resolution required by forefront medical, electronic and nano-technological applications on variably extended (sub-millimeter to centimeter-sized) fields of view. Similarly to what is performed in PALM[46] (Photo-Activated Localization Microscopy) and STORM[47] (STochastic Optical Reconstruction Microscopy), our strategy exploits the automated sub-diffraction centroid localization of sparse temperature increments primed by modulated raster-scanned focused laser light and imaged by a thermal camera. The super-resolution image of the light-absorbing centers in the sample is reconstructed and color-coded by the localized centers and amplitudes of all the measured temperature peaks. Provided the fit localization precision is only limited by the shot-noise of thermal emission[48], the spatial resolution of the rendered image can in principle be tuned down to the ∼1 μm diffraction-limited laser spot size at the excitation visible to near-infrared wavelength. <60-μm resolution is initially demonstrated with temperature increments as low as ∼0.5–5 °C by proof-of-principle experiments on synthetic samples. We thereby achieve 6-fold and 20-fold enhancements with respect to the diffraction-limited prediction and the effective resolution of our thermo-camera in default operation. Validated here and particularly advantageous on a low-cost (∼5k$) thermo-camera, our super-resolved imaging procedure could be adopted to enhance the resolution of any, even expensive (∼70k$), infrared thermography setup. We finally demonstrate application to biological samples, and provide super-resolution maps of the distribution of 30-nm Prussian blue nanocubes[49,50] across millimeter-sized explanted murine skin biopsies. This is a necessary step for the development of a photo-thermal therapy protocol to be subsequently applied in-vivo to the tissue in pathological conditions. Demonstrated here with exogenous nanostructures, exemplary application of super-resolution thermal imaging can be envisioned in the characterization of the melanin distribution in pigmented skin lesions, aimed at the objective discrimination of benign tissue and malignant melanoma both in-vivo and ex-vivo in a non-invasive label-free approach.

## Results

### Photo-activated thermography at sub-diffraction resolution.
The radiance of a gray-body of emissivity $\varepsilon$ ($0 < \varepsilon < 1$) at temperature $T$ is $R = \varepsilon \sigma T^4$, where $\sigma = 5.67 \times 10^{-8}$ Wm$^{-2}$K$^{-4}$ is Stefan-Boltzmann's constant. Non-contact thermal imaging produces thermographic sequences based on the intensity of emitted

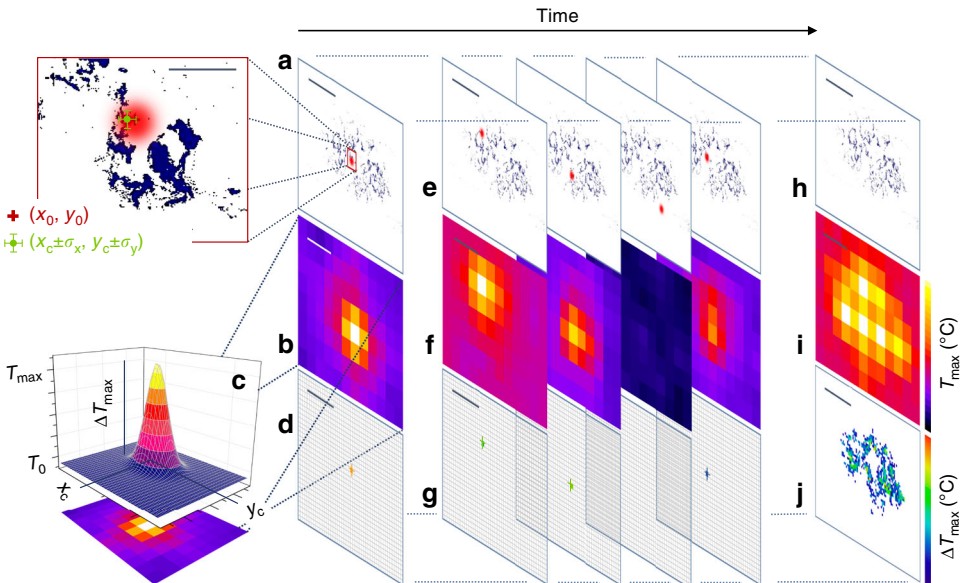

**Fig. 1** Photo-activated thermal imaging at sub-diffraction resolution. A Gaussian laser beam (red spot in **a**) is focused on the sample at $(x_0, y_0)$ with a square pulse of duration $\tau_{on}$. Light absorbing entities (blue) induce a local temperature increase, which is measured by the thermal camera as an evolving 2D Gaussian peak. A Gaussian fit of the frame collected at time $\tau_{on}$ (**b**, **c**) provides the temperature variation $\Delta T_{max} = \Delta T_{max} - T_0$ and the peak coordinates $(x_c \pm \sigma_x, y_c \pm \sigma_y)$ that localize the center of the distribution of the absorbing objects within the laser spot size (magnified region in **a**). The procedure is repeated over sets of isolated points as exemplified in **e**, **f**. The center coordinates provided by the Gaussian fit of temperature peaks identify the position of the absorbing entities on the scan grid (**d**, **g**) with uncertainty $\sigma_{x,y}$ and a color code assigned by the best-fit $\Delta T_{max}$. A maximum projection of the signal across the raw thermo-camera image sequence provides a low ($\sim$mm) resolution image of the sample (**i**), whereas a maximum projection of the stack containing all the localized absorptive centers provides the super-resolution image (**j**) of the scanned region (**h**). Scale bars (100 μm in the magnification of **a**, 1000 μm elsewhere) indicate the typical imaged fields of view; note that the ~50-μm laser spot and the whole magnified area in **a** lie within a single ~400-μm typical thermo-camera pixel on the sample plane. All panels have been derived from experimental data acquired on an explanted murine skin biopsy treated with 30-nm Prussian blue nano-cubes analogous to the one employed for Fig. 3.

radiation detected in the far infrared ($\sim$7–14 μm) wavelength range by an array of microbolometers (thermo-camera)[33–35]. It allows monitoring the sample radiance $R(\mathbf{x}, t)$, and the temperature $T(\mathbf{x}, t)$ provided the emissivity is known, versus time and space at a typical $\sim$10 ms temporal resolution. The spatial resolution, theoretically limited at $\sim$0.3 mm by Abbe's law and the diffraction of far-infrared radiation at the collecting lens, is typically lowered to an effective value $\sim$1 mm by heat diffusion effects.

Active photo-thermal imaging at sub-diffraction resolution is achieved here by sparsely priming heat release in the sample via modulated laser-light illumination, and by subsequently identifying absorptive centers based on the accurate localization of the resulting isolated laser-induced temperature variations. To this aim, a low power Gaussian laser beam, with wavelength lying within the sample absorption band (here, $\lambda_{exc} = 633$ nm), is focused on the sample with beam variance $\omega_0^2$. We consider at first the simple case of a single square-wave excitation pulse of duration $\tau_{on}$, centered at $(x_0, y_0)$. If the laser beam impinges on light absorbing and heat releasing entities (Fig. 1a), an approximately Gaussian 2D temperature distribution is observed in the acquired thermal camera images (Fig. 1b and Supplementary Note 1). The Gaussian amplitude obeys a temporal exponential rise and decay, with the maximum value at time $\tau_{on}$ equal to $\Delta T_{max} = T_{max} - T_0$, where $T_0$ is the equilibrium sample temperature in the absence of illumination; a non-linear 2D Gaussian fit of the temperature peak at time $\tau_{on}$ provides the amplitude $\Delta T_{max}$, together with the Gaussian center coordinates $(x_c, y_c)$ (Fig. 1c, Supplementary Note 1 and Supplementary Figs. 1 and 2). Importantly, $x_c$ and $y_c$ identify the center of the distribution of the laser-excited absorbing objects within the area assigned by the excitation laser spot size. For a given thermal

camera pixel size, the uncertainty $\sigma_{x,y}$ in the determination of the best-fit coordinates $(x_c, y_c)$ is a function of the number of collected infrared photons and can be made arbitrarily small at increasing laser power and temperature variations (see section Calibration and proof-of-principle experiments on synthetic samples). Therefore, localization of absorbing objects can be performed well below the theoretical resolution limit set by diffraction of far-infrared radiation at the thermal camera collecting lens. Achievement of sub-diffraction information by the localization of sparse signal peaks is indeed possible, as demonstrated by well-established techniques including PALM[46], STORM[47], and Single Particle Tracking[51,52].

In order to reconstruct a super-resolution image across extended areas, the procedure of illumination and localization of absorbing centers is sequentially repeated (Fig. 1e). A minimum distance $\Delta \mathbf{x}$ between pairs of consecutively irradiated points and/or a minimum time interval $\Delta t$ in between their illumination have to be adopted to ensure that the two temperature variations appear as spatially and/or temporally separate peaks in the thermal camera images. This is crucial to enable the correct localization of the center of each detected temperature increase by Gaussian fitting. The minimum values for $\Delta \mathbf{x}$ and $\Delta t$ depend on the laser activation time and on the sample thermal diffusivity, as detailed in Supplementary Note 2. Any illumination scheme satisfying this constraint on $\Delta \mathbf{x}$ and $\Delta t$ can be exploited. We adopt here a modulated laser illumination (Supplementary Fig. 3 and Methods section), where the sample is raster scanned by the focused laser beam while a synchronized shutter allows modulating the laser activation and de-activation. The scan grid, $N_x \times N_y$ pixels in format, has a pixel size $\delta x \sim 10$ μm and is oversampled relatively to the $\sim$400-μm thermo-camera pixel size on the sample plane. During each raster-scan,

illumination (open-shutter condition) is only allowed on the limited subset of isolated pixels lying on a square lattice with characteristic spacings $\Delta x$ and $\Delta y$ along the horizontal and vertical directions, respectively (Supplementary Fig. 3). Complete coverage of the investigated area is achieved when the raster scan is repeated $\Delta x \Delta y$ times, since for each scan only $N_x N_y/(\Delta x \Delta y)$ pixels get illuminated. The laser pixel dwell time coincides with the shutter opening time ($\sim$100–1000 ms) during laser activation, whereas a $\sim$2 ms dwell time is selected during shutter closure (see Methods section). This results in a total acquisition time $t_{tot} = N_x N_y \tau_{on} + N_x N_y \tau_{off}(\Delta x \Delta y - 1) \sim$10–100 min over millimeter-sized areas depending on $\tau_{on}$, $N_x$, $N_y$, $\Delta x$, and $\Delta y$ (Supplementary Fig. 3).

During the scanning, the thermal camera acquires frames at rate $f_{rate}$, providing a typical $10^4$–$10^5$-frames sequence (Fig. 1b,f). If a maximum projection of the signal is performed along the time axis of the raw image stack, a low-resolution thermal image—resembling the one that would be produced under wide-field illumination of the sample—is obtained (Fig. 1i and Supplementary Note 3). By contrast, when the center coordinates and amplitudes of all the detected temperature peaks are determined by Gaussian fitting and are associated to the corresponding pixel along the scan grid (Fig. 1d, g), the center coordinates provide much better resolved topological information on the absorptive centers in the sample. Best-fit amplitudes encode the information on the sample absorption and local induced temperature variations, and provide the color code for the final super-resolution image that is obtained by super-imposing all the color-coded pixels where absorptive centers have been localized (Fig. 1j and Methods section). Significantly, if the scan pixel size and the localization uncertainty are smaller than the beam standard deviation ($\delta x$, $\sigma_{x,y} \ll \omega_0$), the spatial resolution of the reconstructed image is ultimately limited by the VIS/NIR excitation spot size which can be focused at will down to the diffraction-limited waist $\sim$1 μm in size. In this regard, it is worth remarking that the visible-wavelength (633 nm) beam is exploited to prime emission of thermal signal at much longer (far infrared) wavelength and is still subject to light diffraction: the proposed imaging approach allows overcoming the diffraction limit for the thermal wavelengths of the detected signal only. This is an important difference with respect to the PALM[46] and STORM[47] techniques, that instead allow super-resolving objects located inside the excitation laser spot.

**Proof-of-principle experiments on synthetic samples**. Custom black ink patterns have been produced by microfiche printing (see Methods section) to provide controlled sub-resolved structures with desired shape and characteristic dimensions in the $\sim$10–100-μm scale. These synthetic samples have been exploited at first to optimize the data analysis procedure and to quantify the uncertainty affecting the localization of temperature peaks.

The theoretically attainable resolution assigned by the excitation spot size is only achieved if the coordinates ($x_c$, $y_c$) retrieved from the Gaussian fit of temperature peaks are determined with an uncertainty $\sigma_{x,y} \ll \omega_0$. In practical cases where $\sigma_{x,y}$ is comparable to $\omega_0$, both contribute in defining the experimental resolution in the reconstructed images.

$\sigma_{x,y}$ depends on the signal-to-noise ratio of the fitted thermal camera images (on the total number of detected photons, which is related to the amplitude of the temperature variation), and on the adopted function for peaks fitting. The optimization of the fitting routine and the quantification of $\sigma_{x,y}$ have been performed on both a uniform and a non-uniform distribution of heat-releasing entities, by periodically priming local absorption at a fixed spatial

location and by evaluating the repeatability in the localization of the $\sim$150 resulting consecutive temperature variations. Symmetric, asymmetric and skewed[53,54] Gaussian functions have been tested (Supplementary Note 4), and measurements have been performed at increasing $\tau_{on} = 0.1$–4 s. Provided $\tau_{on}$ is shorter than the time required to reach a temperature plateau, changing $\tau_{on}$ is equivalent to changing the laser power and allows inducing temperature increases of variable amplitude.

The results obtained on a uniform microfiche ink square are reported in Supplementary Fig. 4. At each $\tau_{on}$, the tested functions equally behave in terms of fit $\chi^2$, peak amplitude $\Delta T_{max}$ and peak coordinates ($x_c$, $y_c$). By contrast, $\sigma_{x,y}$ is systematically minimized when the fit of temperature peaks is performed as a two-step Gaussian fit: a first fit to a symmetric function with fixed variance provides the peak coordinates, which are fixed in a second step to recover the variance $\zeta^2$ and $\Delta T_{max}$. Similarly, a symmetric Gaussian trial function outperforms asymmetric and skewed fit surfaces when the laser beam hits on a microfiche ink stripe with 30-μm width, which only partially covers the excitation spot with $1/e^2$ diameter $56 \pm 2$ μm (Supplementary Fig. 5). Exploitation of the same fitting routine is justified therefore for any a priori unknown distribution of absorptive entities.

Plotting the uncertainty $\sigma_{x,y}$ resulting from the two-step Gaussian fit as a function of the best-fit $\Delta T_{max}$ for both the ink square and the ink stripe further reveals that all data points lie along the expected curve[46,48] $\sigma_{x,y} \approx \sqrt{(\zeta^2 + a^2/12)/N + 4\sqrt{\pi}b^2\zeta^3/(aN^2)} \approx \sqrt{\alpha/\Delta T_{max} + \beta/\Delta T_{max}^2}$. $a$ is the thermo-camera pixel size on the sample plane and $b$ is the number of background photons in the fitted images. $N$ is the number of collected infrared photons, which is directly proportional to the induced temperature increment based on Stefan-Boltzmann's law[33]: $N \propto \varepsilon\sigma(T_{max}^4 - T_0^4) = \varepsilon\sigma((T_0 + \Delta T_{max})^4 - T_0^4) \cong 4\varepsilon\sigma T_0^3 \Delta T_{max}$ (with our typical $T_0 = 293$ K and $\Delta T_{max} = 2$ K the approximation only results in a 1% underestimate of the exact value; the approximation holds within a 5% underestimate up to $\Delta T_{max} = 10$ K). The fit of the trend of $\sigma_{x,y}$ versus $\Delta T_{max}$ (Supplementary Fig. 5e, best-fit parameters $\alpha = 599 \pm 221$ °C μm$^2$ and $\beta = 634 \pm 133$ °C$^2$ μm$^2$) allows extrapolating the localization uncertainty at any measured $\Delta T_{max}$, thereby providing the estimate of $\sigma_{x,y}$ for the following experiments on synthetic samples.

Microfiche samples have been subsequently exploited to perform proof-of-principle experiments aimed at validating the proposed super-resolution imaging approach.

A LABS-shaped structure (acronym for Laboratory of Advanced Bio-Spectroscopy–Fig. 2a) has been reconstructed first. Following experimental measurement of the sample emissivity (Supplementary Note 5), the temperature variations induced by modulated illumination and localized by two-step Gaussian fitting of the thermo-camera frames have been exploited to render the final super-resolution image of the sample (Fig. 2c). Visual inspection of Fig. 2c already suggests proper reconstruction of the features of the printed pattern, and the 0.14 °C standard deviation of the measured temperature variations confirms that the sample is remarkably uniform in terms of the absorption properties (Supplementary Fig. 6). None of the letters of the LABS pattern would instead be recognized by the thermal camera in conventional operation, as can be seen on the temporal maximum projection of the whole stack of the raw thermo-camera images (Fig. 2b).

Quantitatively, the accuracy of the image reconstruction algorithm is demonstrated in Fig. 2d by the comparison with the transmitted-light image of the same sample. The spatial profile of $\Delta T_{max}$ values drawn along the spacing between the

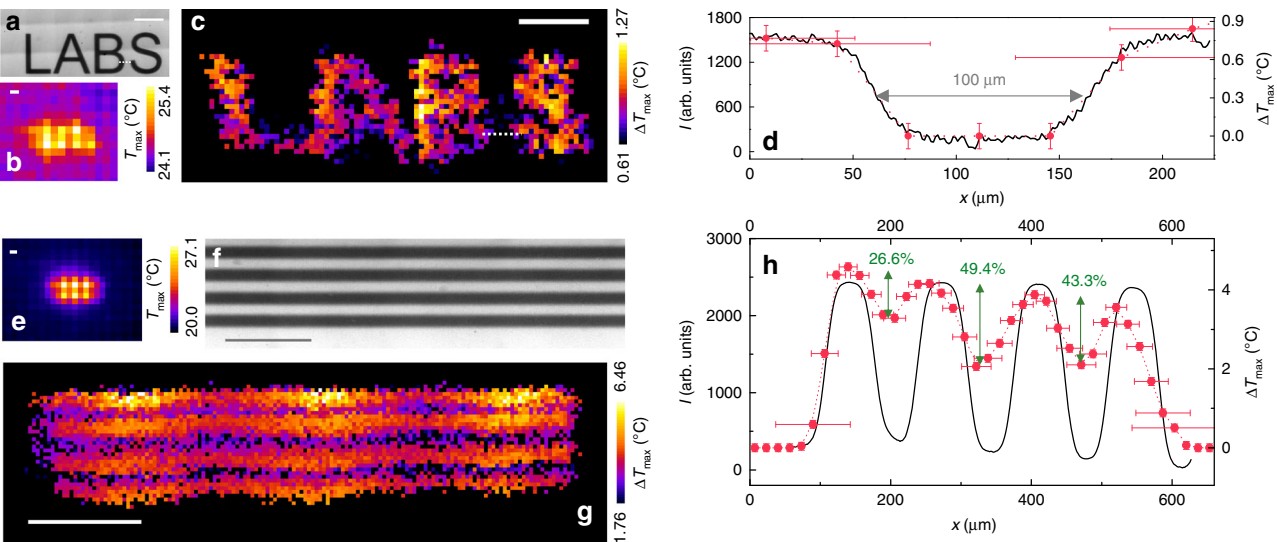

**Fig. 2** Proof-of-principle experiments. **a**, **f** Transmitted-light images of microfiche samples. **b**, **e** Temporal maximum projection of the raw thermo-camera image stacks acquired with modulated illumination on the samples in **a** and **f**, respectively; acquisition parameters: $N_x \times N_y = 120 \times 36$, $\delta x = 34.4$ μm, $\Delta x = 60$, $\Delta y = 6$, $\tau_{on} = 300$ ms, $P = 4.6$ mW, beam $1/e^2$ diameter $56 \pm 2$ μm in **b**, $N_x \times N_y = 160 \times 42$, $\delta x = 15.3$ μm, $\Delta x = 160$, $\Delta y = 6$, $\tau_{on} = 300$ ms, $P = 15$ mW, beam diameter $22 \pm 1$ μm in **e**. **c**, **g** Super-resolution images of the samples in **a** and **f**, obtained from the image stacks employed for **b** and **e**, respectively; $\Delta T_{min} = 0.3$ °C in **c**, 0.7 °C in **g**. **d** Black: intensity profile along the dashed line in **a** upon a lookup-table inversion on the image; red: $\Delta T_{max}$ profile along the dashed line in **c**. **h** Black: average intensity profile along the vertical direction in **f** upon lookup-table inversion; red: average $\Delta T_{max}$ profile along the vertical direction in **g** with contrast percentages between adjacent peaks. In **d**, **h**, uncertainties on $\Delta T_{max}$ values equal the thermo-camera sensitivity 0.1 °C and x-axis error bars for non-zero $\Delta T_{max}$ values are extrapolated from the $\sigma_{x, y}$-versus-$\Delta T_{max}$ plot. Scale bars = 500 μm.

letters B and S of the ink pattern correctly superimposes to the corresponding intensity profile in the transmitted-light image, within the 30–50 μm localization error $\sigma_{x,y}$ expected from the values of $\Delta T_{max} = 0.6$–0.9 °C (Supplementary Fig. 5e). The agreement between temperature variations and transmitted-light intensity profiles should further increase at increasing observed $\Delta T_{max}$ (increasing laser intensity or activation time and decreasing $\sigma_{x,y}$). Still, the localization error $\sigma_{x,y}$ of the reported data-set allows discriminating absorbing structures (in Fig. 2d, the letters B and S) 100 μm apart.

The 100 μm resolution achieved in Fig. 2c, d, which already demonstrates a 3.5-time gain with respect to the theoretical 345-μm diffraction-limited resolution of our thermal imaging setup (Supplementary Note 3), is limited by the $56 \pm 2$ μm laser spot $1/e^2$ diameter (Supplementary Fig. 7), combined with the 30–50 μm localization uncertainty. The resolution has been pushed further by both increasing the excitation laser power (and the detected $\Delta T_{max}$ values) and reducing to $22 \pm 1$ μm the spot $1/e^2$ diameter. A microfiche sample reproducing a grid of uniform ink stripes, 60 μm in width and relative distance (Fig. 2f), has been imaged by modulated illumination yielding the super-resolution image in Fig. 2g. Again, the structures that would be unresolved in conventional thermal imaging (as shown by the temporal maximum projection over the raw image stack, Fig. 2e) appear instead discernible in the super-resolved frame. The comparison with the transmitted-light (200-nm resolution) image of the same sample quantitatively confirms that the size of the imaged ink stripes is correctly retrieved (Fig. 2h). Furthermore, the average spatial profile of temperature increments detected orthogonally to the ink stripes satisfies the 26.4% contrast threshold required by the well-known Rayleigh's criterion[55], thereby quantifying in ≤60 μm the achieved resolution (Fig. 2h). This corresponds to a 5.8-time gain relatively to the diffraction-limited prediction, and a 20-time gain with respect to the $(1200 \pm 180)$-μm thermo-camera resolution in conventional operation (Supplementary Note 3 and Supplementary Fig. 8).

**Sub-diffraction thermal imaging on biological samples.** Sub-diffraction thermal imaging has been finally tested on biological systems with sparse distributions of visible-light absorbers (Fig. 3). Explanted murine skin biopsies have been treated with 30-nm Prussian blue nanocubes (PB-NPs), which exhibit a strong absorption band centered at 700 nm and a highly efficient thermal relaxation upon VIS/nearIR irradiation[49,50]. After experimental characterization of the sample emissivity and of its heterogeneity (Supplementary Note 5 and Supplementary Fig. 9), the temperature increments induced by modulated illumination and localized with the two-step Gaussian fitting procedure have been exploited to provide a super-resolution spatial map of the distribution of the nanostructures inside the tissue (Fig. 3b). Details as close as ~40 μm appear resolved under the adopted imaging conditions, as highlighted by the magnified area in Fig. 3c. It is to be noted that the lateral size of the magnified region corresponds to the thermo-camera theoretical 345-μm diffraction-limited resolution. The achieved resolution gain appears even larger when the reconstructed images (Fig. 3b, c) are compared to the conventional thermographic image obtained as the temporal maximum projection over the raw thermo-camera image stack (Fig. 3d).

A high degree of co-localization has been observed between the temperature variations in the super-resolution image and the lowest-intensity pixels in the corresponding transmitted-light image of the same sample, ascribed to PB-NPs (Fig. 3a–c and Supplementary Fig. 10). This highlights the specificity of thermal detection and proves the possibility of imaging NPs at sub-diffraction resolution in a heterogeneous environment. This conclusion is reinforced by the results obtained on nanoparticle-untreated samples under identical imaging conditions (Fig. 3f–h). In the absence of exogenous nanoparticles, nearly no temperature variation is detected. Only if the $\Delta T_{min}$ threshold is lowered from 0.3 °C (as adopted for treated samples in Fig. 3b) to 0.1 °C (corresponding to the thermal camera sensitivity), a few spurious temperature increases get identified (Fig. 3g and Supplementary

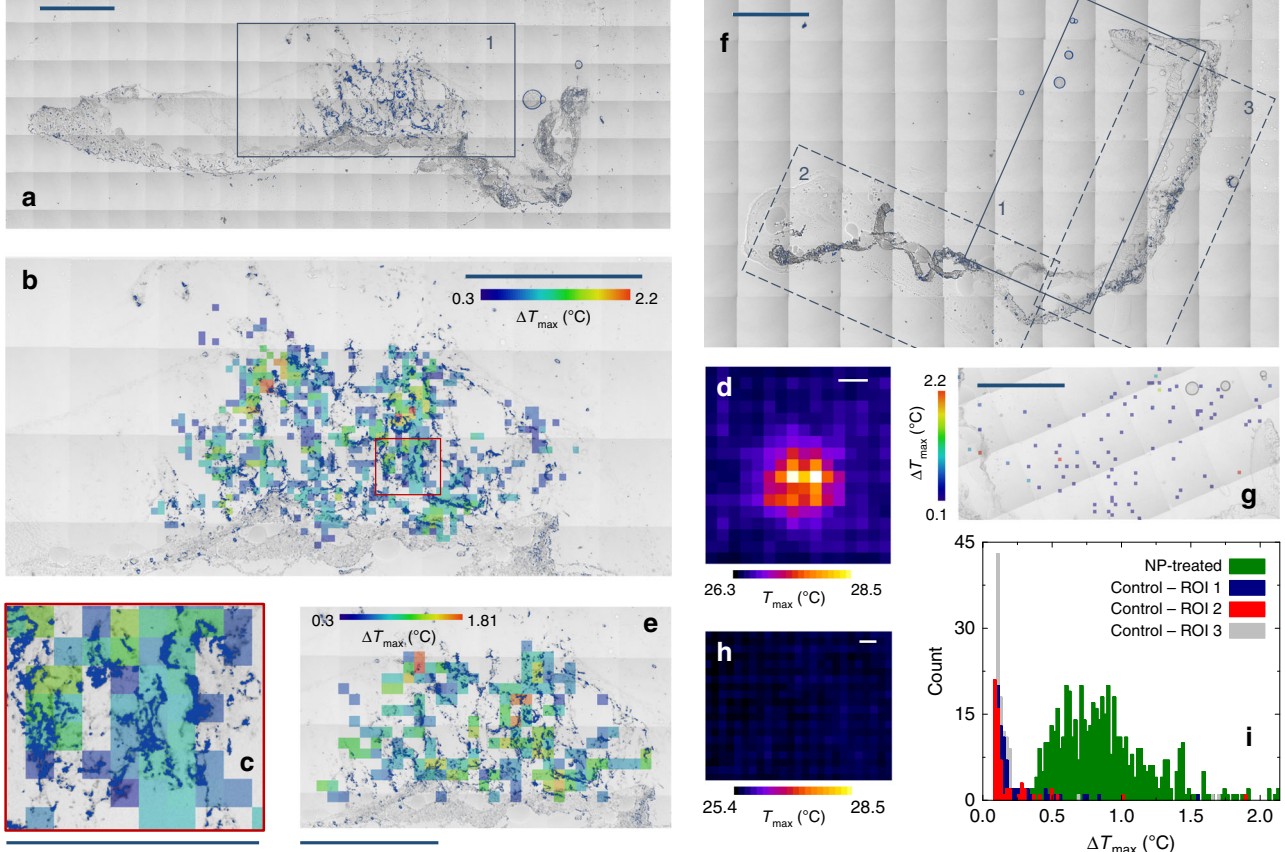

**Fig. 3** Sub-diffraction thermal imaging on murine biopsies. **a** Transmitted-light tile-scan image of an explanted murine skin biopsy treated with 30-nm PB nanocubes, which are highlighted in navy by an upper threshold on intensity counts. **b** Super-resolution image of ROI 1 in **a** ($N_x$x$N_y$ = 100 × 48, $\delta x$ = 37.6 μm, $\Delta x$ = 50, $\Delta y$ = 1, $\tau_{on}$ = 1 s, P = 15 mW, beam $1/e^2$ diameter 22 ± 1 μm, $\Delta T_{min}$ = 0.3 °C) overlaid to the transmitted-light image (nanocubes in navy as in **a**). **c** Magnification of the red boxed region in **b**. **d** Temporal maximum projection of the thermo-camera stack employed for **b**. **e** Same as **b** with doubled pixel size ($N_x$x$N_y$ = 50 × 24, $\delta x$ = 75.3 μm, $\Delta x$ = 25, $\Delta y$ = 2), overlaid to the transmitted-light image (nanocubes highlighted in navy as in **a**); only a 2.7 × 1.6 mm² ROI is shown for the sake of display. **f** Transmitted-light tile-scan image of a nanoparticle-untreated explanted murine skin biopsy; pixels are highlighted in navy by the intensity upper threshold as in **a**. **g** Super-resolution image of ROI 1 in **f** (same acquisition parameters of **b**, $\Delta T_{min}$ = 0.1 °C); analogous results have been obtained in ROIs 2 and 3 in **f** (Supplementary Fig. 11). **h** Temporal maximum projection of the thermo-camera stack employed for **g**. **i** Histograms of the $\Delta T_{max}$ values of the super-resolution images of ROI1 in **a** and ROIs 1–3 in **f**. Scale bar = 345 μm in **c**, 1 mm elsewhere.

Fig. 11). Comparison of the histograms of the $\Delta T_{max}$ values in treated and untreated samples allows indeed to unambiguously ascribe the induced heat release in treated samples to injected nanoparticles (Fig. 3i).

We finally remark that $\Delta T_{max}$ values as low as 0.3–1.5 °C provide sufficient signal-to-noise ratio to enable the discrimination of NP clusters ~40 μm apart (Fig. 3c). The corresponding laser power $P$ = 15 mW is sufficiently low not to induce any photo-damage of the imaged structures, as verified through sequential imaging by transmission microscopy before and after photo-thermal data acquisitions (Supplementary Fig. 12). At fixed laser power and illumination time, the total acquisition time varies with the scan parameters $N_x$, $N_y$, and $\delta x$. A doubled pixel size allows retaining the major information on the PB-NPs distribution inside the treated tissue (Fig. 3e), while reducing the total acquisition time from 90 (Fig. 3b) to 20 min over the 3.7 × 1.8 mm² scanned area.

## Discussion

We have proposed a photo-modulated thermal imaging technique that we have showed to overcome Abbe's resolution limit for the emitted thermal wave on a conventional thermo-camera. We take advantage of a compact, easily aligned and low-cost

(hardware cost ~20k€) benchtop microscope, where a continuous-wave laser beam is raster-scanned and synchronously modulated by a programmable shutter. By relying on the localization of the isolated induced temperature variations, our approach reconstructs the distribution of the absorptive centers in the sample and combines this morphological information with the quantitative measurement of induced temperature variations.

Sub-diffraction 60-μm resolution has been demonstrated by proof-of-principle experiments, thereby enabling a 5.8-enhancement with respect to the 345-μm diffraction-limited prediction and a 20-time enhancement with respect to the 1200 ± 180 μm effective resolution of our thermal camera in conventional operation. The achieved resolution already proves sufficient to accurately map the distribution of light-absorbing nanostructures injected into explanted murine skin biopsies.

We have shown that, with single-spot scanning, a 3.7 × 1.8 mm² area can be scanned with 40-μm pixel size in 90 min, and this time reduces to 20 min with a 75-μm scan pixel size. Extended acquisition times are common to all the super-resolution techniques (e.g., PALM and STORM) that rely on a stochastic molecular switching and readout[46,47,56]. A number of bio-relevant applications would benefit from the achievement of super-resolution spatial information while not necessarily requiring transient thermal imaging at high temporal resolution:

we can envision exemplary applications for our imaging approach in the characterization of the distribution of metallic nanoparticles in explanted biopsies for the optimization of photo-thermal therapies, or in the development of active thermography ex-vivo pre-clinical protocols in the context of melanoma screening and diagnosis. Whenever beneficial for in-vivo biological applications, our current imaging time can also be shortened by multi-spot illumination (via a Spatial Light Modulator) and/or by relaxing the constraint on the observation of individual, isolated temperature peaks. Classes of algorithms allow handling multiple and overlapping signal peaks, based on deep-learning, sequential fitting, sparsity, or maximum-likelihood estimation, and could be adopted to speed up the imaging process[57–59]. Methods developed for ghost imaging[60] could be borrowed as well, especially for those applications (e.g., in cryptography) that require the recognition of an encrypted but known pattern.

By taking advantage of the full analytical description of the space-time dependence of the photo-activated temperature spots imaged on the thermal camera (Supplementary Note 1), and by properly treating both the sample transparency to infrared wavelengths and its heterogeneity in terms of emissivity and thermal diffusivity (Supplementary Note 5), we should also be able to combine the reconstruction of super-resolved images to the quantitative characterization of the sample thermal properties. The possibility of producing 2D maps of (spatially heterogeneous) diffusivity values is currently being investigated. Similarly, efforts are being directed toward the measurement of local nanoparticle concentrations starting from measured maps of laser-primed temperature increments.

## Methods

**Optical setup.** Supplementary Fig. 3a shows a schematic of the experimental photo-thermal imaging setup. Sample absorption is primed by a He-Ne laser beam ($\lambda_{exc} = 633$ nm, Thorlabs, NJ, USA) with 30-mW maximum output power. Gray filters are employed to adjust the laser power on the sample plane and achieve a minimum 0.1 °C temperature variation with a relatively short illumination time $\tau_{on}$ ~10–1000 ms. The excitation spot size on the sample plane may be adjusted by a two-lens Keplerian beam reducer with focal lengths $f_1$ and $f_2$: the $1/e^2$ beam diameter ranges from $56 \pm 2$ μm in the absence of the beam reducer down to $22 \pm 1$ μm with $f_1 = 3$ cm and $f_2 = 10$ cm (Supplementary Fig. 7).

A scanning unit with servo-board electronics and a pair of galvanometric mirrors (MicroMax Series 670, Cambridge Technology Inc., MA, USA) is coupled to a two-lens scan system ($f_3 = 4$ cm, $f_4 = 10$ cm), followed by beam focalization ($f_5 = 10$ cm) onto the sample plane. The mirror positioning system is driven by an Arduino Uno microcontroller board and interfaced with a custom written Python code. Mirrors are operated in conventional raster-scanning mode (typical pixel size, 10–100 μm on the sample plane), with user-defined voltage values supplied to the mirrors to regulate the scan path length along the horizontal and vertical axes. The voltage dependence of the path length has been quantified by following the calibration procedure described with Supplementary Note 6 and Supplementary Fig. 13.

The Arduino board and Python code used to drive the scan system are also exploited to synchronize the galvanometric mirrors with the electronic shutter (Oriel 76992, Newport, CA, USA; 150 Hz maximum operation frequency) aimed at modulating the laser illumination in time. During laser de-activation (i.e., shutter closure), a default 2-ms pixel dwell time of the laser beam along the scan grid is usually adopted. During laser activation, the shutter opening time is set to the desired value $\tau_{on}$ and the laser pixel dwell time is temporarily increased to the same value to ensure the sample illumination occurs at the same location. Since neither the thermal relaxation time nor the sample characteristic heating time is significantly affected by the sample thermal diffusivity (Supplementary Fig. 1a), a simple practical criterion can be adopted to choose $\tau_{on}$ irrespectively of the sample thermal properties: based on the desired temperature peaks amplitude (i.e., the desired signal-to-noise ratio, which will impact in turn on the peaks localization uncertainty and the achievable resolution), we adopt the minimum laser illumination time and the maximum laser power that ensure the achievement of such a desired signal amplitude while avoiding any sample photo-damage. Once the laser activation time has been set, the minimum spatial distance between pairs of consecutive illumination events is specified according to the sample thermal properties following the criteria we detail in Supplementary Note 2.

The detection of thermal radiation is performed by an uncooled microbolometer-based thermal camera (FLIR E40, FLIR Systems Inc., OR, USA) with 30-Hz frame rate. The detector provides $320 \times 240$ images, with $25° \times 19°$ FOV and ~400 μm pixel size on the sample plane. As detailed with the measurement procedure in Supplementary Note 5, the thermo-camera automatically corrects for the atmospheric attenuation and the sample reflection of the thermal radiation emitted by the surroundings[33]. To this aim, the sample distance $d$ from the camera front lens, the sample emissivity $\varepsilon$, and the relative humidity $h$ and temperature of the atmosphere have to be provided to the camera software. $h = 50\%$ has been assumed for all the experiments of the present work, whereas the sample distance $d$ (~30 cm) and the ambient temperature have been measured for each imaging experiment. The emissivity has been experimentally characterized by the black-tape method[33] for both microfiche ink samples ($\varepsilon = 0.8$) and biological specimens ($\varepsilon = 0.95$) as described in Supplementary Note 5. Finally, imaging of a known-size object at constant temperature has been exploited before each experiment to calibrate the pixel size and take small variations associated to the thermo-camera positioning into account.

All the experiments have been performed with a maximum 20° vertical tilt of the thermal camera with respect to the sample (Supplementary Fig. 14). Such a thermal camera tilt does not impact on measured $\Delta T_{max}$ values (Supplementary Note 5), nor it impacts on the size of imaged objects, the foreshortening effect of perspective view being excluded by the experimental results reported in Supplementary Fig. 15.

**Transmission microscopy setup.** A Leica TCS SP5 STED-CW scanning confocal microscope (Leica Microsystems, D) has been employed for transmitted-light imaging. The laser source consists in a 633 nm He-Ne beam (power $P \sim 10$ μW on the sample plane), which is focused on the sample by a $20 \times 0.5$-N.A. air objective (HCX PL Fluotar, Leica Microsystems, D). Images have been acquired by detecting the transmitted light signal with a non-spectral dedicated photo-multiplier tube, with no confocal pinhole along the detection optical path. A 400-Hz raster scan frequency per line has been adopted, and millimeter-sized sample regions have been imaged using a tile-scan acquisition mode.

**Prussian blue nanoparticles.** PB nanoparticles have been prepared according to the synthesis protocol reported in refs. [49,50]. starting from 10 mM FeCl$_3$ (Fe$^{III}$), 10 mM K$_4$[Fe(CN)$_6$] (Fe$^{II}$) and 0.025 M citric acid reagents. Solutions have been heated to 60 °C during the synthesis and then allowed to cool at room temperature. Purification has been performed by ultracentrifugation for 25 min at 13,000 rpm, followed by pellet re-suspension in half the original volume.

The synthesis leads to nanocrystals with cubic shape and an average side of $(29 \pm 8)$ nm, measured by TEM imaging. The absorption band peaks at 700 nm, with a relative 60% absorbance at the 633-nm wavelength we employ for photo-thermal imaging.

**Ink samples.** Microfiche printing has been employed to produce reference ink patterns with convenient shape and known sub-resolved size for the initial experimental demonstration of super-resolution photo-thermal imaging. Patterns have been designed (PowerPoint, Microsoft Office) as white shapes over a black background (microfiche printing produces the negative of the original drawing). The pattern linear dimensions have been selected based on the desired size on the final printed film, taking the 28-times reduction factor of the printing procedure into account. Imaging by conventional transmission microscopy at 633 nm allows inspecting the uniformity of the printed ink layers as in Fig. 2a, f.

**Biopsies.** Murine skin biopsies have been explanted from the flank derma of sacrificed Balb/c mice at 6 weeks of age. Explanted tissue sections have been embedded in OCT freezing medium (Biooptica, I), cut in 10-μm sections on a cryostat and adhered to glass slides (Superfrost Plus, Thermo Fisher Scientific, MA, USA) for photo-thermal imaging. When necessary, Prussian blue nanoparticles (100 μL, 10× concentration of the initial stock) have been injected into the biopsies right after the explant. All the experiments have been performed under protocols approved by the Institutional Animal Care and Use Committee of the University of Milano-Bicocca and the Italian Ministry of Health.

**Image reconstruction.** Raw data for super-resolution thermography consist in an image sequence of ~$10^4$–$10^5$ frames acquired by the thermal camera during the modulated laser illumination of the sample. Due to the ~$10 \times 8$ cm$^2$ field of view of the thermal camera employed here, the analysis is restricted to a Region Of Interest (ROI) covering the scanned area and easily identified by computing the maximum projection over the time axis of the acquired image stack. For each pixel $i$ within the ROI, the temperature-versus-time profile $T_i(t)$ may contain N temperature peaks with approximately exponential rise and decay. Each peak is produced by a laser square pulse impinging on light-absorbing and heat-releasing entities located within the $i$-th pixel or in pixels nearby (due to heat diffusion). Since the scan pixel size $\delta x$ oversamples the thermal camera pixel $\delta x_T$ ($\delta x \sim 10$–100 μm, $\delta x_T \sim 400$ μm), tens to hundreds of peaks can be detected in a single $T_i(t)$ plot; peak amplitudes vary according to the distance of the illuminated heat-releasing objects from pixel $i$ and from the center (highest-intensity portion) of the excitation laser spot at the time of illumination. A pattern-recognition algorithm can be exploited to locate the position of each peak on the time axis of the $T_i(t)$ profile. In the present case, fits to a 1D symmetric Gaussian function (that reasonably approximates the exponential rise and decay for the short adopted $\tau_{on} = 0.3$–1 s) are performed by sliding a fit

window over the entire time trace. The temporal coordinate of each local temperature maximum is stored, and the corresponding frame of the thermo-camera image stack is fitted to a Gaussian surface for the sub-diffraction localization of the center coordinates $(x_c, y_c)$ of the laser-induced temperature increase.

Once the procedure has been repeated for all the pixels in the ROI, the peak coordinates $(x_c, y_c)$ and amplitudes $\Delta T_{max}$ of all the identified temperature increases are exploited to reconstruct the super-resolution image of the scanned area. Since the pixel size in the final image coincides with the pixel size $\delta x$ of the scan grid, a bin size $\delta x$ is employed to group all the best-fit peak coordinates $(x_c, y_c)$ into the corresponding pixels along the scan grid. Then the highest among the $\Delta T_{max}$ values associated to the temperature variations contained in each pixel is selected to provide the color-code for the pixel itself. A minimum threshold on rendered temperature increments (referred to as $\Delta T_{min}$ in the text) is generally employed to exclude very low and spurious temperature variations that might be erroneously detected by the peak identification algorithm even in the absence of light absorption on noisy frames. The threshold does not eliminate any true laser-induced temperature variation, since the typical $\Delta T_{max}$ values are at least twice as high as $\Delta T_{min}$. We finally remark that the rendered image can be reported either with the amplitude of temperature variations (as in the present case), or with the absolute values obtained by summing temperature variations to the baseline temperature in the absence of laser illumination.

**Data analysis software**. Photo-thermal imaging raw data have been acquired with the thermo-camera software (FLIR Tools +, FLIR Systems Inc., OR, USA), exported in .csv file format and entirely processed by a custom written Python code. Both non-linear 1D fits of temperature time profiles and the symmetric, asymmetric and skewed Gaussian surface fits of detected 2D temperature peaks have been performed by the same code with the curve fitting routine of the Scipy open source Python tool. Reconstructed super-resolution images have been saved as.txt files and rendered with ImageJ (US National Institutes of Health, MD, USA) for visualization and look-up table adjustment. Their comparison with the results of transmitted-light imaging has been accomplished by the Origin Pro 8 software (Origin Lab Corporation, MA, USA) based on temperature and intensity profile plots.

## Data availability
The datasets generated and analyzed in the present study are available from the corresponding author on reasonable request.

## Code availability
The custom code employed for the analysis of the datasets acquired in this study is available from the corresponding author on reasonable request.

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

## Acknowledgements

We acknowledge the funding from Università degli Studi di Milano-Bicocca for the year 2018.

## Author contributions

Ma.B. designed the experiments, analyzed the data, and wrote the paper. Ma.B. and M.M. acquired the data, conceived, and wrote the data-analysis code. A.Z., M.M., and Ma.B. built and tested the experimental setup. M.C. and G.C. conceived the technique, supervised the project and wrote the paper. P.P. and My.B. provided Prussian blue nanoparticles. F.M. and F.G. prepared and provided biological samples. L.S. helped with biological samples and related data analysis. L.D. wrote the paper and discussed the project.

## Competing interests

The authors declare no competing interests.
