## [Peer Review File · Nature Communications]

Reviewers' comments:

Reviewer #1 (Remarks to the Author):

This work aims to provide an approach to obtain sub-diffraction thermal image, by 2D fitting of the isolated temperature in the thermo-camera frames. I think the author would agree that this approach is a derivative of the well-known PALM and STORM.

This is still publishable if there could have clear and direct implications. In that regard, a few issues need to be addressed.

1. The work is trying to make biological and/biomedical relevance. One degreeC is actually not a small number in biological samples. Then the resolution it can achieve could still rather limited.

Please comment on that.

2. Another requirements for this technique is the absorber. What are the requirements for the absorbers? In this study, 30nm Prussian blue nanocubes are used. What are the limitations?

Reviewer #2 (Remarks to the Author):

NCOMMS-10-04263

This manuscript reports a “super-resolution” scanning photothermal imaging technique. This technique is based on a far-field optics, realized by (1) locally heating a light-absorbing sample with a laser (633 in wavelength); (2) taking a thermal image of the sample with a IR thermal camera, and (3) identifying a peak amplitude and its position based on 2-D Gaussian fitting of the IR thermal image from the camera. Based on the Stefan-Boltzmann law with the grey assumption of $\varepsilon = 0.95$, the identified amplitude is converted to temperature. The authors obtained a super-resolution temperature mapping by raster-scanning the laser spot and repeating the thermal imaging and Gaussian fitting.

The reported work is technically sound and may realize a fast, large-area photothermal imaging with a high resolution that cannot be obtained by commercial far-field thermal cameras. The authors provide proof-of-principle measurements for micropatterned black ink samples and an explanted murine skin sample treated with 30-nm Prussian blue nanocubes (PB-NPs). However, the reviewer cannot recommend this manuscript for publication in Nature Communications, in large part due to the authors’ overclaim without clear explanations of physics behind the technique. Without addressing points that the reviewer is concerned of, this manuscript only reports an imaging processing technique, which has been used in other optical microscopy techniques, first applied to thermal imaging.

Major Concerns:

1. The title and introduction are confusing and may seriously mislead the readers. First of all, the reviewer is seriously concerned of the authors’ repeated using the term “sub-diffraction”. Let me make it clear. The presented work is not sub-diffraction imaging by any means, fundamentally and practically. The obtained results may beat the resolution of a commercial thermal camera, but still subject to the fundamental diffraction limit. The imaging processing based Gaussian fitting does not convey any sub-diffraction physical information – it is merely a mapping technique by picking up a peak value of diffraction-limited thermal image for a certain pixel. Furthermore, the image resolution that the authors achieved was $\sim 60 \mu\text{m}$, which is still bigger than the characteristic wavelength of thermal radiation at the elevated sample temperature as the authors mention in line 49. Then, what is the rationale of the authors’ claim that their imaging method is a sub-diffraction technique? Their claim of sub-diffraction gives the reviewer impression that the authors either do not understand the fundamental aspects of optics or overclaimed their technique by overlooking the physical meaning of sub-diffraction, both of which do not qualify as a rigorous research article.

2. The introduction emphasizes the motivation and the potential impacts of the presented imaging technique in the biomedical field, such as thermal imaging of blood circulation and metabolism, cellular pathogenesis of diseases, and cell stimulation and differentiation induced by endogenous or exogenous temperature variations. However, it should be noted that most of these events are self-heating and thus should not have external heating stimuli for adequate imaging. The presented imaging is not a pure thermal imaging (i.e., temperature mapping without external heating, such as SThM or fluorescence-based thermal imaging) that measures a sample temperature change resulting from internal heat generation, but a sort of pump-probe imaging that provides the thermal

responses upon laser heating – therefore reveals optical and thermophysical properties of a sample. The authors should clearly state how the presented work can be used for the aforementioned biomedical imaging needs. Otherwise, the introduction should be written to adequately address the potential impact of the presented work. In addition, in lines 43-45, the authors state that no micro-absorption images covering $100 \times 100 \mu\text{m}^2$ with sub- μm resolution has not been reported. It should be noted that it is not because a large-area scanning was not possible but because the previous works focused on nanoscale thermal imaging. In addition, the presented work also cannot achieve a sub- sub- μm resolution. Many imaging techniques aim either large area or a high resolution, but do not tackle the two aims with one technique unless complimenting with two techniques. The presented work is no exception – to the reviewer, this work aims more onto a large-area thermal imaging than achieving a high resolution. Again, although the obtained result may be better than the resolution of a conventional thermal camera, it is way worse than other cutting-edge nanoscale thermal imaging techniques.

3. The presented technique realizes a high-resolution thermal image by only considering the peak value of the Gaussian fitted temperature elevation. This technique may work for homogeneous samples as tested in this work, but the reviewer is not convinced if it is applicable to heterogenous samples – what would be the contrast limits in thermal diffusivity and/or emissivity of sample components that can be resolved with the proposed imaging technique? Moreover, a substrate should play a critical role in heat diffusion and thermal emission processes – the thermal images are basically convoluted images between sample and substrate’s thermal and radiative properties. For example, if the thermal diffusivity of the substrate is much higher than that of a sample or the sample is semi-transparent in the IR range, the obtained image would not provide a discernible image of the sample and would be extremely challenging to decouple the sample information from the background signal from the substrate. The limit of the proposed technique in this regard should be carefully examined and discussed. A simple assumption of grey emissivity of 0.95 for all samples is very weak.

Besides the major concerns, the reviewer also found specific comments/questions that should be addressed by the authors.

Specific Comments:

1. In lines 38-39, the authors describe one of the drawbacks using fluorescence-based thermometry: “drawbacks are the impact of the chemical properties of the surrounding medium on the fluorescence signal”. However, the authors have to treat their biological sample with Prussian Blue nanoparticles (PBNP) to receive any noticeable thermometry signal. Can the authors comment on how treating a sample with PBNPs is beneficially different than treating a sample with fluorescent dyes?

2. The authors cite scanning thermal microscopy and fluorescence imaging as techniques for high resolution thermometry, yet they leave off scanning thermorefectance microscopy. While scanning thermorefectance microscopy is not a sub-diffraction technique, its use of a visible probing laser enables it to readily obtain spatial resolutions $\sim 1 \mu\text{m}$. It should therefore see an acknowledgement in the literature review of this paper with key citations.

3. Can the authors comment on how they considered the view factor in their determination of the surface temperature? It is unclear how the Stefan-Boltzmann law can be used to directly interpret surface temperature when then it is important to determine how many photons are incident on the camera. Simply saying in the Methods that they included the camera-sample distance in the camera software, may mislead readers into thinking they did not consider the amount of radiation emitted by the sample and received by the camera.

4. What is the viewing angle with respect to the sample? Can the authors comment on whether consideration of this will affect their results? This citation describes how the viewing angle has dramatic effects on the measured temperature signal: Ball et al. “Factors affecting the accuracy of thermal imaging cameras in volcanology”, *Journal of Geophysical Research*, Vol. 111, B11203, 2006.

5. The sentence in line 127 – 129 is not clearly written.

6. All figure captions are not easy to comprehend with too many subfigure numbers. They could be clearer with elaborations.

7. Can the authors comment on how they optimized their laser spot size? They discuss in detail how smaller spot sizes can improve the imaging resolution, but do not comment on the potential drawbacks of reducing the laser spot size. Sample damage? Low signal?

8. In line 198 and other places, the authors specify the thickness of the ink stripe as 30 μm . However, its should be the width, not thickness. This should be corrected. With this pointed out, what is the thickness of the ink stripe? This should be thick enough to exclude the substrate emission effect.

9. The authors should specify the criteria for the approximation in line 207.

10. Questions about Figure 2:

- How do we trust the temperature scale on the secondary y-axes and image color scales? To strengthen the author’s claim that they are quantitatively measuring temperature, they should provide at least finite element calculations to elucidate the theoretical temperature rise expected under the laser illumination conditions. This will verify their assumption on the emissivity of the material and their usage of the Stefan-Boltzmann law.
- The line traces in (d)-(f) and (k)-(l) compare the transmission optical images with the obtained temperature maps. Why are the low temperature data points (i.e., off of the ink features) all at exactly zero? The authors provide y-scale uncertainties for each data point; however, the average value of these data points do not vary across the line traces within these uncertainties? Moreover, why is there no x,y -scale uncertainty shown for the low temperature data points?
- The authors select seemingly random lines from the temperature images for the line traces in (d)-(f) and (k)-(l). It appears that only temperature line traces that match the best with the optical line traces are selected. While the temperature image is somewhat able to discern the “LABS” text, it is not as highly resolved as the optical image. However, the selected line traces may mislead in that the temperature resolution is comparable to the optical image resolution.
- The authors claim that the spatial resolution of their thermal image is $\sim 60 \mu\text{m}$ or smaller. If this is true, Fig. (i)-(i) should clearly distinguish the line patterns separated $60 \mu\text{m}$ apart.

However, they do not clearly distinguish line patterns. The reviewer is not convinced of the authors' claimed spatial resolution. More careful experiments should be conducted to rigorously define the best spatial resolution of their method, like imaging line patterns with different separations until the obtained thermal images cannot clearly distinguish the line patterns.

- Using two color scale bars in one figure is confusing. The authors should select one.
- Using different scale bars without texts in figures is confusing.

11. Questions about Figure 3:

- Is it necessary to show boxes 2 and 3 in the images of Fig. 3(a) and (f), or remove boxes 2 and 3 from (a)? The results of these boxes are not shown in the main text but may confuse a reader who may be looking for those results.
- In particular, why have the authors not included the results of boxes 2 and 3 from Fig. 3(f)? They include the results of box 1 into Fig. 3(g), but this section has the least biological sample. Can the authors explain why this choice was made?
- In addition to the previous comment, the authors compare the histograms of (g) and (b) in Fig. (i) to illustrate the role of the PBNP in increasing the temperature rise of the biological sample; however, there is almost no biological sample in (g). Is this really a fair comparison or should the result of either box 2 or 3 in (f) be used?
- The authors should provide more physical discussion about the observed temperature distribution in Fig. (b) and (c). Why does the temperature map vary as it does? It is difficult to understand why certain areas are purple while others are light green. The reviewer suspects that this difference might be due to the different distribution of PB-NPs, but there may be effect from the uneven sample thickness and emissivity distribution. No explanations are given from the authors although this seems to be a very important question.
- Clarification question: Are regions that are not attributed a color below a delta temperature of 0.3 K?
- How can (g) be used to evaluate the sample emissivity if there are no PBNP? Can the authors clarify that the PBNP are very transparent in the emission window 7-14 μm ?
- How do the authors accurately determine the emissivity of each of their sample materials? Can they justify why selecting 0.95 is appropriate for each experiment?

Simple Comments:

1. Line 161 – error (broken font) in the parentheses
2. The paragraphs on p. 10 are unnecessarily short leading to choppy reading.
3. In line 203, “lye” seems like a typo of “lie”.
4. Lines 222-226 contain redundant statements. Can the authors make this discussion more concise?
5. Line 357: It is unclear what is meant by “reducing the resolution twice”. Technically, making the resolution smaller by two times results in a fourfold increase of the number of pixels and resulting acquisition time.

Answers to the comments of Reviewer #1.

REVIEWER'S COMMENT: *This work aims to provide an approach to obtain sub-diffraction thermal image, by 2D fitting of the isolated temperature in the thermo-camera frames. I think the author would agree that this approach is a derivative of the well-known PALM and STORM.*

ANSWER: We agree with the Reviewer. These methodologies, cited in the original manuscript (lines 61-65), indeed inspired our work. The main similarity of our approach with these techniques consists in achieving super-resolved information by the a-posteriori centroid localization of signal peaks. This very same image processing procedure based on Gaussian peak fitting is not only peculiar to PALM and STORM, but it is also exploited, for example, by the well-established Single Particle Tracking (SPT) in fluorescence microscopy to reconstruct particle trajectories with sub-diffraction accuracy (Ruthardt, N. et al. *Single-particle tracking as a quantitative microscopy-based approach to unravel cell entry mechanisms of viruses and pharmaceutical nanoparticles*, Mol. Ther., 19, 7, 1199-1211, 2011 - Yildiz, A. et al. *Myosin V walks hand-over-hand: single fluorophore imaging with 1.5-nm localization*, Science, 300, 2003).

At the same time, there are major differences between our approach and PALM/STORM. First, image reconstruction in our case is not relying on the use of an extrinsic probe specifically designed to be photo-activatable; instead, photo-thermal imaging can be used with any molecule that is absorbing in the sample under examination (endogenous or exogenous, as discussed in the answer to the following comments) provided it allows efficient heat release upon laser light absorption. Secondly, PALM and STORM allow super-resolving objects located *inside* the excitation laser spot, and beat the fundamental diffraction limit at the visible wavelengths of both the excitation beams and emitted fluorescence signal. On the contrary, we overcome the diffraction limit for the far infrared wavelengths of the detected thermal signal only. The visible-wavelength (633 nm) beam is still subject to light diffraction, and the excitation laser spot size assigns a limit to the spatial resolution that we can achieve.

REVIEWER'S COMMENT (cont'd): *This is still publishable if there could have clear and direct implications. In that regard, a few issues need to be addressed. 1. The work is trying to make biological and/biomedical relevance. One degree C is actually not a small number in biological samples. Then the resolution it can achieve could still rather limited. Please comment on that.*

ANSWER: We agree with the Reviewer that inducing a thermal increase might stress significantly a biological sample. However, the relevance and consequence of the stress also depend upon the particular application and/or system under study. For example, several clinical applications of diathermy (Tecar® therapy or laser heating for muscle contraction and pain relief) use temperature increases much higher than 1°C, currently operating at 40°C (Draper, D.O., *Temperature change in human muscle during and after pulsed short-wave diathermy*, J. Orthop. Sports Phys. Ther., 29(1):13-22, 1999). Other fields of application of externally induced temperature variations deal with the discrimination of healthy tissues from malignant skin lesions in vivo. In these cases, a temperature increase of few Celsius degrees does not cause significant discomfort to the patient (Okabe, T. et al., *First-in-human clinical study of novel technique to diagnose malignant melanoma via thermal conductivity measurements*, Sci. Rep. 9:3853, 2019 - Herman, C. *The role of dynamic infrared imaging in melanoma diagnosis*, Expert Rev. Dermatol. 8(2):177-184, 2013). Equivalently, a one-degree temperature variation does not typically affect or permanently damage the tissue state when applied to explanted biopsies (see for example Supplementary Figure 9 of our initial submission). Regarding living cells applications, cellular apoptosis can be induced at temperature higher than 42°C, at least 5°C above the physiological 37°C. Several papers in the literature in the field of neuronal regeneration suggest that a controlled temperature increase of several degrees can promote cell proliferation and growth (Kudo, T. et al. *Induction of Neurite Outgrowth in PC12 Cells Treated with Temperature-Controlled Repeated Thermal Stimulation*, PLOS ONE, 10(4), 2015 - Paviolo, C. et al., *Laser Exposure of Gold Nanorods Can Increase Neuronal Cell Outgrowth*, Biotechnol. Bioeng. 110, 2013). Overall, the literature provides plenty of examples of biological systems where a temperature stimulus of the order of (or even well above) 1°C can be easily tolerated.

Along with these considerations, whenever thermal stress has to be minimized, we can also adapt the image acquisition parameters for super-resolution thermal imaging by bearing in mind how the induced temperature increase correlates with the signal/noise ratio of the acquired images. Here, the typical temperature noise on the bolometer is about 0.1 °C as stated by the thermal-camera producer. This means that we can lower our

requirements on the temperature increase from 1 °C to 0.4 °C still having a signal/noise ratio of 4 and a corresponding localization accuracy $\sim 70 \mu\text{m}$ (Supplementary Fig.5e in the revised submission). This is still sufficient to significantly outperform the $\sim 350 \mu\text{m}$ resolution of the Germanium optics of the thermal camera in conventional operation while reducing the thermal stress on the sample.

REVIEWER'S COMMENT (cont'd): *Another requirements for this technique is the absorber. What are the requirements for the absorbers? In this study, 30nm Prussian blue nanocubes are used. What are the limitations?*

ANSWER: We thank the Reviewer for this interesting question. Our method can be extended to any endogenous or exogenous compound in the sample that is able to efficiently release heat upon proper excitation (in these regards our method could also be implemented on a multiwavelength basis, for example by using a battery of excitation diode lasers). In the present work, we used PB nanoparticles that we are currently developing for photo-thermal therapy. These nanoparticles are biocompatible, absorb around 700 nm and release heat very efficiently. Mapping their spatial distribution with tens-of-microns accuracy while preserving the information on the induced temperature increase (as in Fig.3) is an important step for the subsequent development and calibration of a photo-thermal therapy protocol to be applied in-vivo to the tissue. A similar future application is the detection of metallic nanoparticles in biopsies of different organs in order to characterize the biodistribution after systemic administration in animal model systems, in an alternative way with respect to two-photon luminescence or transmission electron microscopy (Meng, F. et al., *Quantitative assessment of nanoparticle biodistribution by fluorescence imaging, revisited*, ACS Nano, 12(7):6458-6468, 2018). Regarding applications with endogenous absorbers, imaging protocols have been recently developed in the context of confocal photo-thermal microscopy to discriminate lesions from healthy skin tissue based on the spatial distribution of melanin aggregates in melanoma model mice (Kobayashi, T. et al. *Label-free imaging of melanoma with confocal photothermal microscopy: differentiation between malignant and benign tissue*, Bioeng. 5, 67, 2018). Melanin is endowed with a large absorption cross-section for visible light without substantial fluorescence emission, resulting in efficient heat-release mechanisms and providing a useful marker for the identification of (the histologic transition between) dysplastic nevi, in situ melanoma and malignant melanoma in photo-thermal imaging. Recent research suggests one compelling application area of active thermal imaging is dermatology, and highlights that the application of our protocol to skin lesions where the absorber is melanin is a natural breakthrough of this starting work (Kobayashi, T. et al. *Label-free imaging of melanoma with confocal photothermal microscopy: differentiation between malignant and benign tissue*, Bioeng. 5, 67, 2018 - Okabe, T. et al., *First-in-human clinical study of novel technique to diagnose malignant melanoma via thermal conductivity measurements*, Sci. Rep. 9:3853, 2019 - Herman, C. *The role of dynamic infrared imaging in melanoma diagnosis*, Expert Rev. Dermatol. 8(2):177-184, 2013). Following the Reviewer's concern, considerations aimed at clarifying the potential implication as well as biological/biomedical relevance of the present work have been added to the manuscript introduction.

Answers to the comments of Reviewer #2.

GENERAL COMMENT: *The reported work is technically sound and may realize a fast, large-area photothermal imaging with a high resolution that cannot be obtained by commercial far-field thermal cameras. The authors provide proof-of-principle measurements for micropatterned black ink samples and an explanted murine skin sample treated with 30-nm Prussian blue nanocubes (PB-NPs). However, the reviewer cannot recommend this manuscript for publication in Nature Communications, in large part due to the authors' overclaim without clear explanations of physics behind the technique. Without addressing points that the reviewer is concerned of, this manuscript only reports an imaging processing technique, which has been used in other optical microscopy techniques, first applied to thermal imaging.*

ANSWER: We disagree with the Reviewer in claiming that no clear explanation and quantitative discussion of the physics behind our technique was included in the manuscript. In this regard, consider please at least Supplementary Notes 1.1 and 1.3, lines 83-87 and lines 205-207 of the main text, and Supplementary Figs. 1, 3 and 4 of the original submission.

Before entering a specific analysis of these parts (see next paragraph), we would like to point out that our approach is a photo-activated super-resolution thermal imaging that relies on the automated localization of sparse thermal loads induced by the sample absorption of modulated raster-scanned focused laser light. As it is clarified in the original manuscript, light-absorbing centers get identified by the 2D fit of temperature peaks appearing in the acquired thermal camera frames. Also in view of the well-established photo-activation super-resolution techniques in fluorescence microscopy (PALM and STORM – refs. 31 and 32 in the original Ms.), these considerations determine the physical bases of our approach in terms of: (i) the heat equation under modulated laser illumination, (ii) the Stefan-Boltzmann's law, and (iii) the Abbe limit. Indeed, the solution of the heat equation assigns the spatial and temporal evolution of the sample temperature during both laser illumination and thermal relaxation. The Stefan-Boltzmann's law is at the basis of the sample thermal emission and of the infrared signal detected in the thermal camera frames. Abbe's limit assigns the theoretical diffraction-limited resolution of our thermal camera (and the diffraction-limited size of the excitation visible laser spot), as thoroughly discussed in the answers to the Reviewer's major comments (see Major Comments #1, #2 and #3, below).

Coming back to the analysis of the original Ms., we would like to raise the Reviewer's attention on the fact that both Abbe's and Stefan-Boltzmann's equations were included in the original Ms. and Supplementary Material, along with a description of the space and time dependence of the amplitude and variance of temperature variations primed by pulsed laser illumination. Related discussions in the original submission are summarized hereafter:

- In Supplementary Note 1.3, we discussed the resolution of conventional diffraction-limited thermal imaging. In doing so, we analyzed the limitations theoretically enforced by Abbe's law on the thermal camera optics, and we explicitly quantified the diffraction resolution limit provided the numerical aperture and spectral range of the thermal camera employed for all the experiments in the present work.
- We referred to Stefan-Boltzmann's law in the manuscript main text at lines 83-87. We also exploited it at lines 205-207 to relate the number of collected infrared photons to the amplitude of laser-induced temperature variations, and thereby provided a physical description of the temperature dependence of the peaks localization uncertainty (reported in Supplementary Fig.4e).
- In Supplementary Note 1.1 and Supplementary Fig. 1, 3 and 4, we demonstrated that a 2D Gaussian properly approximates the temperature distribution primed by the sample absorption of a focused Gaussian laser pulse and imaged on the thermal camera frames. We proved that a non-linear surface fit or the acquired images allows accurate determination of the peaks amplitude and center coordinates, which are the only parameters needed for image reconstruction in the proposed super-resolution imaging approach. Our considerations were based on exemplifying experimental data and the rationale behind this choice was justified explicitly in Supplementary Note 1.1: we stated there that the analytical solution of the heat equation under modulated laser illumination can only be achieved in the time interval corresponding to laser activation ($[0, \tau_{on}]$), whereas the theoretical space and time dependence of the induced temperature increase have to be derived numerically during thermal relaxation (at time $t > \tau_{on}$).

We think the Reviewer would acknowledge that the essential theoretical bases of our method were already included in the original submission. However, based on the Reviewer’s comment, we have now added the aforementioned analytical and numerical treatment of the heat equation. Given the issue of the limited space in the main text, the treatment is added to the Supplementary Material (Supplementary Note 1.1 and Supplementary Fig.1 in the revised submission) and mentioned in the Ms. (section “*photo-activated thermography at sub-diffraction resolution*”). This further discussion of the physical bases of our work would be valuable in paving the way to a more extensive exploitation of our photo-induced thermal imaging approach for the measurement of the thermal diffusivity of the sample. At the same time, by highlighting the explicit dependence of the measured temperature signals on the material properties, the theoretical treatment we have added to Supplementary Note 1.1 allows a deeper investigation of how the sample thermal diffusivity affects the signal/noise ratio and applicability of our approach. We will also refer to the solution of the heat equation in our answer to the Reviewer’s major comment #3.

MAJOR COMMENT #1: *The title and introduction are confusing and may seriously mislead the readers. First of all, the reviewer is seriously concerned of the authors’ repeated using the term “sub-diffraction”. Let me make it clear. The presented work is not sub-diffraction imaging by any means, fundamentally and practically. The obtained results may beat the resolution of a commercial thermal camera, but still subject to the fundamental diffraction limit.*

ANSWER: We firmly disagree with the Reviewer on this point.

The resolution limit enforced by light diffraction at the thermal camera collecting lens is predicted by the well-known Abbe’s law, stating that two objects emitting radiation at wavelength λ and imaged by collecting optics with numerical aperture N.A. can only be resolved if separated by a minimum distance

$$\Delta r \approx 0.61 \frac{\lambda}{N.A.} \quad (1)$$

The Germanium optics employed in thermal imaging systems typically reach very low numerical apertures. As we computed and reported in the original Supplementary Note 1.3, the f-number and focal distance of the thermo-camera lens employed here, operating at 30 cm from the sample plane, result in a numerical aperture N.A.=0.023. Correspondingly, with a thermal wavelength $\lambda=13 \mu\text{m}$, a $345 \mu\text{m}$ resolution limit is predicted by Abbe’s law. In Figure 2 we demonstrated imaging of the emitted thermal power at $60 \mu\text{m}$ resolution: this fully justifies our claimed sub-diffraction resolution (please note Fig. 2 has been revised as discussed in the answer to the Reviewer’s specific comments) for the thermal radiation emitted by the sample.

In this regard and for the sake of clarity, we also highlight an important difference with respect to the PALM and STORM techniques. PALM and STORM allow super-resolving objects located *inside* the excitation laser spot, thereby beating the fundamental diffraction limit at the visible wavelengths of both the excitation beams and emitted fluorescence signal. In our case, the visible-wavelength (633 nm) beam is still subject to light diffraction and is exploited to prime emission of a thermal signal at much longer (far infrared) wavelength: the proposed imaging approach allows overcoming the diffraction limit for the thermal wavelengths of the detected signal. This has been clarified in the revised Ms. We also would like to point out that our technique is a super-resolution (i.e., beyond the diffraction limit, like in PALM) imaging of thermal radiation, but it is not (as the Reviewer seems to have meant) a sub-wavelength imaging technique (such as SNOM, for example) with respect to this radiation.

MAJOR COMMENT #1 (cont’d): *The imaging processing based Gaussian fitting does not convey any sub-diffraction physical information – it is merely a mapping technique by picking up a peak value of diffraction-limited thermal image for a certain pixel.*

ANSWER: We firmly disagree with the Reviewer. The data say the contrary (Figs. 2 and 3) and demonstrate that the resolution in the reconstruction of the spatial distribution of the temperature increases in the sample can be as low as $60 \mu\text{m}$ (please see the original and updated Fig.2), well below the theoretical diffraction limit for the far infrared thermal wavelengths detected with a commercial thermo-camera. As now strengthened in the revised Ms., achieving sub-diffraction resolution by the localization of signal peaks is the same kind of approach of the PALM and STORM methods for super-resolution microscopy in the visible (please, see also

our discussion of your “General Comment” above). It is also worth remarking that the same image processing procedure, based on Gaussian fitting and peak identification, is also exploited by Single Particle Tracking (SPT) for the reconstruction of particle trajectories in fluorescence microscopy: starting from diffraction-limited fluorescence images, the coordinates of the imaged particles get determined with sub-diffraction \sim nm accuracy based on the surface fit of the corresponding intensity peak (Ruthardt, N. et al. *Single-particle tracking as a quantitative microscopy-based approach to unravel cell entry mechanisms of viruses and pharmaceutical nanoparticles*, Mol. Ther., 19, 7, 1199-1211, 2011 - Yildiz, A. et al. *Myosin V walks hand-over-hand: single fluorophore imaging with 1.5-nm localization*, Science, 300, 2003). A reference to SPT techniques and related literature has been added to the revised text (“Results” section).

MAJOR COMMENT #1 (cont’d): *Furthermore, the image resolution that the authors achieved was $\sim 60 \mu\text{m}$, which is still bigger than the characteristic wavelength of thermal radiation at the elevated sample temperature as the authors mention in line 49. Then, what is the rationale of the authors’ claim that their imaging method is a sub-diffraction technique? Their claim of sub-diffraction gives the reviewer impression that the authors either do not understand the fundamental aspects of optics or overclaimed their technique by overlooking the physical meaning of sub-diffraction, both of which do not qualify as a rigorous research article.*

ANSWER: In my modest opinion, here it is not a matter of “not understanding the fundamental aspects of optics”, as the Reviewer states and that I instead modestly but firmly claim for myself. The optical resolution limit is known in undergraduate courses of physical optics and I think it is not fair to doubt about the knowledge of this principle by a colleague (at least not in the first instance). Apart from these considerations, that I regret I had to do, the objection and the claim of the Reviewer may be due to a confusion between “sub-wavelength” and “sub-diffraction”.

Sub-diffraction techniques in visible optics reach spatial resolution of few tens of nanometers, four to five times less than the typical Abbe’s diffraction-limited resolution that, for the high numerical aperture optics available for the visible radiation, is of the order of 200-300 nm. This value is already below the visible radiation wavelength because of the high numerical aperture of the optics. In thermal imaging, due to the low numerical aperture available, Abbe’s limit corresponds to a spatial resolution (in our case $\cong 345 \mu\text{m}$) larger than the thermal radiation wavelength. The limit reached by us here, $\cong 60 \mu\text{m}$, is indeed a super-resolution result, but not a sub-wavelength condition.

We hope that our aim, the physical basis of our approach and its value are now clear. To summarize our answers to the Reviewer’s major objection:

- The physical bases of our approach are the Abbe’s and the Stefan-Boltzmann’s laws, along with the heat equation under modulated laser illumination. All were already cited in the original Ms., and further analytical and theoretical treatment is now included in the Supplementary Material;
- The approach consists in a super-resolution photo-induced thermal imaging: it beats Abbe’s resolution limit for the thermal radiation wavelength (not the visible wavelength), and it is not a sub-wavelength imaging;
- Our approach has been inspired by the seminal works on PALM and STORM microscopies and differs from them in: (i) the possible exploitation of the micro-absorption intrinsic to the sample, and (ii) the achievement of super-resolution with respect to the limit set by diffraction of far infrared thermal wavelengths.

MAJOR COMMENT #2: *The introduction emphasizes the motivation and the potential impacts of the presented imaging technique in the biomedical field, such as thermal imaging of blood circulation and metabolism, cellular pathogenesis of diseases, and cell stimulation and differentiation induced by endogenous or exogenous temperature variations. However, it should be noted that most of these events are self-heating and thus should not have external heating stimuli for adequate imaging. The presented imaging is not a pure thermal imaging (i.e., temperature mapping without external heating, such as SThM or fluorescence-based thermal imaging) that measures a sample temperature change resulting from internal heat generation, but a sort of pump-probe imaging that provides the thermal responses upon laser heating – therefore reveals optical and thermo-physical properties of a sample. The authors should clearly state how the presented work can be*

used for the aforementioned biomedical imaging needs. Otherwise, the introduction should be written to adequately address the potential impact of the presented work.

ANSWER: As the Reviewer correctly remarks, our technique is a sort of pump-and-probe technique and can be identified as “photo-activated thermal imaging”, as also stated in the Ms. title. Indeed, it produces super-resolved spatial maps of absorbing centers in the sample by taking advantage of externally induced temperature variations. As a consequence, we definitely agree that our approach does not allow to improve the spatial resolution of the temperature increases arising from internal heat sources in biological samples. We apologize for not discussing it clearly in the original Introduction.

However, a number of biologically and biomedically relevant issues can be tackled with, or even require, “active” thermography, that measures the temperature response when heating (or cooling) is applied to enhance or induce thermal contrast. For example, active-thermography clinical protocols have been developed in the context of melanoma screening and diagnosis to differentiate benign and malignant skin lesions according to either the skin thermal conductivity, or its thermal recovery behavior after thermo-stimulation such as heating or cooling (Okabe, T. et al., *First-in-human clinical study of novel technique to diagnose malignant melanoma via thermal conductivity measurements*, *Sci. Rep.* 9:3853, 2019- Herman, C. *The role of dynamic infrared imaging in melanoma diagnosis*, *Expert Rev. Dermatol.* 8(2):177–184, 2013). The differences in thermal recovery are caused by the increased metabolism and blood perfusion of cancerous lesions, but external thermal stimulus is beneficial in getting them revealed. Furthermore, discrimination between lesions and healthy skin tissue has been performed by photo-thermal (PHI) microscopy in explanted skin sections of melanoma model mice, based on the spatial distribution of melanin aggregates (Kobayashi, T. et al. *Label-free imaging of melanoma with confocal photothermal microscopy: differentiation between malignant and benign tissue*, *Bioeng.* 5, 67, 2018; He. J. et al., *Noninvasive, label-free, three-dimensional imaging of melanoma with confocal photothermal microscopy: differentiate malignant melanoma from benign tumor tissue*, *Sci. Rep.* 6:30209, 2016). Indeed, melanin exhibits advantageously high absorption cross-section for visible laser light and heat-release efficiency, and its absorption spectra and spatial distribution represent well-known markers to differentiate in-situ and/or malignant melanoma from its potential precursors, including melanocytic and dysplastic nevi (Zonios, G. et al., *Melanin absorption spectroscopy: new method for noninvasive skin investigation and melanoma detection*, *J. Biomed. Opt.*, 13(1), 2008 - Kobayashi, T. et al. *Label-free imaging of melanoma with confocal photothermal microscopy: differentiation between malignant and benign tissue*, *Bioeng.* 5, 67, 2018). Therefore, super-resolved photo-activated thermal imaging could be exploited in a label-free approach to spatially map the melanin distribution over relatively large (mm²) areas both in vivo and in explanted skin biopsies. It is also worth remarking that morphological information is already provided by the reconstructed super-resolution images (as shown by Figures 2 and 3), whereas information on the sample thermal diffusivity could be derived by further exploiting the temperature relaxation kinetics included in the same data-sets: the combination of morphological and thermal characterization would enable robust and objective dermatological classification of pigmented skin lesions, which is recognized in the literature as a compelling research area for quantitative dynamic infrared imaging (Herman, C. *The role of dynamic infrared imaging in melanoma diagnosis*, *Expert Rev. Dermatol.* 8(2):177–184, 2013).

Along with melanoma screening and diagnosis, exemplary applications of super-resolution photo-activated thermal imaging can be envisioned in nanomedicine and nanotechnology. As demonstrated in the present work with Fig. 3, the distribution of metallic (non-fluorescent) nanoparticles – that could not be accurately mapped by conventional active infrared imaging - can be reconstructed at high resolution in explanted murine tissue sections. This could be extended to map the distribution of heat-releasing nanoparticles in biopsies of different organs in order to characterize their biodistribution after systemic administration in animal model systems. Such a characterization is of paramount importance in developing protocols for the subsequent drug-delivery or photo-thermal therapy applications of the nanostructures. In these regards, the possibility of combining morphological information with the measurement and calibration (after the correction for the emissivity and possible IR transparency of the sample) of laser-induced temperature variations is a significant advantage of our technique.

The biological and biomedical relevance of the present work, along the lines discussed above, was not commented in detail in our initial submission. We thank the Reviewer for giving us the opportunity to improve the Introduction: it is now largely modified with the inclusion of the literature cited above and hopefully meets the relevant Reviewer’s comments.

MAJOR COMMENT #2 (cont'd): *In addition, in lines 43-45, the authors state that no microabsorption images covering 100'100 μm^2 with sub- μm resolution has not been reported. It should be noted that it is not because a large-area scanning was not possible but because the previous works focused on nanoscale thermal imaging. In addition, the presented work also cannot achieve a sub- sub- μm resolution. Many imaging techniques aim either large area or a high resolution, but do not tackle the two aims with one technique unless complimenting with two techniques. The presented work is no exception – to the reviewer, this work aims more onto a large-area thermal imaging than achieving a high resolution. Again, although the obtained result may be better than the resolution of a conventional thermal camera, it is way worse than other cutting-edge nanoscale thermal imaging techniques.*

ANSWER: We apologize with the Reviewer in case our sentences were not sufficiently clear. The sentence at lines 43-45 was intended to evidence that micro-absorption images acquired by PHI imaging typically cover 100x100 μm^2 fields of view with sub- μm resolution, but do not allow a point-by-point quantitative measurement of the temperature increases (or thermal conductivity values). This observation aside, we fully agree with the Reviewer that any imaging technique has intrinsic limitations in simultaneously improving resolution and field of view. When compared to other cutting-edge nanoscale thermal imaging techniques, our approach:

- allows a spatial resolution in the 10-100 μm range and finds its best application over fields of view hundreds-to-thousands of micrometers in side. This spatial scale is not tackled by imaging techniques aiming at reaching the nanoscale that limit themselves at small fields of view, nor by conventional infrared thermography that lacks the necessary resolution.
- allows simpler and less expensive implementation, with the simplification especially involving the hardware and optical setup.
- allows in principle the measurement of the amplitude of induced temperature variations, provided the emissivity of the sample is taken into account. The transparency of the substrate does not affect appreciably our localization procedure, as discussed in the later section “Major Comment #3, cont'd”. We notice that sample-dependent calibration procedures are also required by thermo-reflectance microscopy to convert the measured signal into a temperature value (Farzaneh, M. et al, *CCD-based thermoreflectance microscopy: principles and applications*, J. Phys. D: Appl. Phys. 42, 143001, 2009), whereas PHI imaging indirectly relies on the index of refraction and does not provide direct estimates of temperature variations.

The description of the key features of our approach has now been modified in the manuscript Introduction according to the Reviewer’s comment.

MAJOR COMMENT #3: *The presented technique realizes a high-resolution thermal image by only considering the peak value of the Gaussian fitted temperature elevation. This technique may work for homogeneous samples as tested in this work, but the reviewer is not convinced if it is applicable to heterogenous samples – what would be the contrast limits in thermal diffusivity and/or emissivity of sample components that can be resolved with the proposed imaging technique?*

ANSWER: We thank the Reviewer for raising this interesting question. We would like at first to point out that the proposed imaging technique exploits the sample heterogeneity arising from the spatial distribution of light micro-absorbers. The temperature variations are imaged only at those positions in the sample where the visible excitation laser beam is absorbed and a temperature variation (above a minimum threshold ΔT_{min}) is produced. Micro-absorbers embedded in patches of the sample with different thermal diffusivity or characterized by different emissivity values, but producing a temperature increase larger than ΔT_{min} , would be included in the reconstructed images.

Resolving in space different thermal diffusivity or emissivity values would not be immediately feasible within the present algorithm, nor was the actual goal of this work. In our present algorithm, the effect of the possible heterogeneity of the sample thermal diffusivity D and emissivity ε can be on: (a) the accuracy in the measurement of ΔT_{max} values, and (b) the variability of the signal/noise ratio over different regions on the acquired thermal camera frames, that in turn regulates the peaks localization uncertainty and the achieved resolution.

Regarding point (a), the accuracy in the measurement of temperature increases from the Stefan-Boltzmann law is affected only by the emissivity heterogeneity, that would result in apparent point-by-point variations in the ΔT_{max} values. For the data reported in Figure 3 we have now explicitly quantified this effect in the answer to

the Reviewer's specific comment #11. We expect no effect on the accuracy in the measurement of the temperature increments from the heterogeneities in thermal diffusivity: the presence of different D values would result in temperature peaks with different amplitudes (as now remarked in the theoretical treatment of Supplementary Note 1.1), that would be correctly revealed by the peak fitting procedure. Obviously, the thermal diffusivity would instead affect the interpretation that we could give of the temperature increases in terms of the micro-absorbers' concentration.

Regarding point (b), since the localization accuracy decreases when the emitted power decreases (Supplementary Figure 5e of the revised Ms.), the sample heterogeneity (both in terms of thermal diffusivity and emissivity) will affect the spatial resolution of the image. If the sample is highly heterogenous, we expect to find patches in the sample where we can reconstruct the distribution of the absorbers with an accuracy higher than in other patches. However, from Supplementary Figure 5e, we see that at $\Delta T_{max} \cong 1^\circ\text{C}$ a 10% change in the measured temperature variation only implies a 7% change of the localization accuracy. Lowering to $\Delta T_{max} \cong 0.5^\circ\text{C}$, the same 10% change in the measured temperature variation would produce a 9% variation in the position accuracy. This variability in the localization precision, arising from variations of the diffusivity or emissivity from points to points of the sample, would smooth some details in the reconstructed images without affecting sensibly the overall morphological reconstruction of the sample.

To completely address the Reviewer's concern, we finally quantify the range of thermal diffusivity of the sample on which our algorithm can still provide us with a map of the absorbers. To this aim, we should remark that:

- The maximum temperature increase under laser illumination is inversely proportional to the sample thermal diffusivity (as now evidenced in the general treatment of the heat equation in Supplementary Note 1.1 and in Supplementary Fig.1);
- The minimum signal/noise ratio for which we can fit a Gaussian peak with an uncertainty in the position of $\cong \frac{1}{3}$ of the nominal spatial resolution of the thermocamera is $\cong 2$ (Supplementary Fig.4e in the initial submission – now Supplementary Fig.5e).

We take as a reference the situation reported in Fig.2c, that corresponds to a laser spot intensity of 0.6 kW/cm^2 , an emissivity $\varepsilon=0.8$ (measured on the microfiche samples as discussed in the Reviewer's specific comments) and a diffusivity $D = 5 \times 10^5 \text{ } \mu\text{m}^2/\text{s}$ (measured with a modification of the laser flash method - Cernuschi, F. et al., *In-plane thermal diffusivity evaluation by infrared thermography*, Rev. Sci. Instrum. 72, 10, 2001). This situation corresponds to an average signal/noise ratio $\frac{S}{N} \cong 9$. Since we can detect by means of the Gaussian peak fitting the presence of absorbers down to a $S/N = 2$ with a spatial resolution of about $100 \text{ } \mu\text{m}$, we can say that the maximum thermal diffusivity of the sample for which we can recover a distribution map, at the excitation intensity used here, is $D_{T,max} \cong 23.5 \times 10^5 \text{ } \mu\text{m}^2/\text{s}$. This value can be increased by raising the laser intensity, provided that there is no photo-damage of the sample. With the samples used here, the intensity can be easily raised by four times without damaging the sample (Fig.2g in its revised form), increasing the maximum value of the diffusivity of the same ratio up to $D_{T,max} \cong 9 \times 10^6 \text{ } \mu\text{m}^2/\text{s}$. On the lower limit, the minimum thermal diffusivity of the sample on which we can work is set by the relaxation time of the temperature jump compared to the line scanning rate: by increasing the time interval in between consecutive illumination events, we can tackle low sample diffusivities with the only drawback of the increased total data acquisition time. We conclude therefore our method can be applied to any thermal diffusivity value in the broad range 10^4 - $10^7 \text{ } \mu\text{m}^2/\text{s}$, covering practically any material from PVC to steel.

MAJOR COMMENT #3 (cont'd): *Moreover, a substrate should play a critical role in heat diffusion and thermal emission processes – the thermal images are basically convoluted images between sample and substrate's thermal and radiative properties. For example, if the thermal diffusivity of the substrate is much higher than that of a sample or the sample is semi-transparent in the IR range, the obtained image would not provide a discernible image of the sample and would be extremely challenging to decouple the sample information from the background signal from the substrate. The limit of the proposed technique in this regard should be carefully examined and discussed.*

ANSWER: We agree with the Reviewer that we must take carefully into consideration the limitations to our method due to (a) the thermal diffusivity of the substrate, and (b) the possible sample semi-transparency in the long wave infrared range.

Both these contributions from the underlying substrate will affect the maximum value of the temperature at the peak. The diffusivity of the substrate reduces the temperature increment on the sample draining heat away. However, unless the substrate has very high thermal diffusivity, the maximum value ΔT_{max} will be identified by the peak fitting algorithm as long as it is above the detection threshold, ΔT_{min} , chosen in the reconstruction software. To this regard, consider please the estimates on the range of accessible thermal diffusivity of the sample: similar considerations can be drawn here for the substrate.

Instead, the semi-transparency of the sample in the far-IR range would produce an undesired bias in the measured temperature variation, that in principle needs for a correction. In fact, what could happen is that part of the heat dissipated by the sample heats the substrate up and the substrate thermal emission is then collected by the bolometer through the (semi-transparent) sample. The correction to the measured temperature increment would then depend on the emissivity of the substrate: more photons than expected are collected, especially when the emissivity of the substrate is higher than the emissivity of the absorbing sample layer.

However, this indirect effect can be mitigated by a direct measure (for example, by the black-tape method) of the emissivity of the sample laying on the substrate. In fact, if the sample is semitransparent to the thermal radiation, we are actually measuring an effective emissivity of the sample+substrate structure (the photon contributions of both the sample and the substrate would be simultaneously taken into account when determining the emissivity value).

We do not expect instead an effect of the thermal diffusivity or the transparency of the substrate on the accuracy with which we recover the peak position of the temperature distribution. The temperature distribution of the substrate imaged through the (partially transparent) sample layer would only widen the approximately Gaussian background below the sharper peak due to the sample absorption: it will not shift its peak position. We expect therefore that our peak center localization algorithm will work as well in the presence of a semi-transparent sample overlaid to a substrate with higher thermal diffusivity.

All the considerations raised by the Reviewer's major comment have been added to the Supplementary Material (Supplementary Note 4).

MAJOR COMMENT #3 (cont'd): *A simple assumption of grey emissivity of 0.95 for all samples is very weak.*

ANSWER: We apologize for not including emissivity measurements in our initial submission, and we thank the Reviewer for highlighting this evident mistake from our side. Emissivity measurements are now reported in Supplementary Note 4 and Supplementary Fig.14 for both synthetic (microfiche) and biological samples, and the assumption of an emissivity of 0.95 for skin sections has been justified. We refer to our answers to the Reviewer's specific comments #4, 10 and 11 for a detailed discussion of the emissivity measurement procedures and of the effect of emissivity heterogeneities in the samples used in the present work.

SPECIFIC COMMENTS:

1. In lines 38-39, the authors describe one of the drawbacks using fluorescence-based thermometry: "drawbacks are the impact of the chemical properties of the surrounding medium on the fluorescence signal". However, the authors have to treat their biological sample with Prussian Blue nanoparticles (PBNP) to receive any noticeable thermometry signal. Can the authors comment on how treating a sample with PBNPs is beneficially different than treating a sample with fluorescent dyes?

ANSWER: We apologize with the reviewer since the reported sentence was unclear. The sentence in the manuscript introduction was aimed at remarking that, even though fluorescence-based thermometry relies on fluorescence parameters (intensity, anisotropy or lifetime) to derive temperature values, these parameters are not affected only by temperature. Bleaching, fluctuations in the illumination intensity and the medium optical or chemical properties (e.g., pH and ionic strength) have an impact on the fluorescence emission, that is hard to correct and therefore affects the accuracy of the technique. The drawback of fluorescence-based thermometry does not consist therefore in the exploitation of exogenous fluorescent dyes, but consists in taking advantage of signal variations that may be due to parameters other than temperature (Donner S. et al. *Mapping intracellular temperature using green fluorescent proteins*, *Nanolett.* 12:2107-2111, 2012 – Okabe K. et al. *Intracellular temperature mapping with a fluorescent polymeric thermometer and fluorescence lifetime imaging microscopy*, *Nat. Comm.* 3:705, 2012).

Regarding PBNPs, treatment with Prussian blue nanocubes should not be detrimental to the sample provided the documented high biocompatibility of the nanoparticles (we remark Prussian blue products received US Food and Drug Administration approval – Dacarro, G. et al. *Prussian blue nanoparticles as a versatile photothermal tool*, *Molecules*, 23:1414, 2018). That being said, PBNPs treatment is not strictly necessary to perform super-resolution thermal imaging. As pointed out in detail in our answer to the Reviewer’s major comment #2, depending on the imaged tissue laser-light absorption and heat release by endogenous absorbing entities can be exploited as well. The same principle is at the basis of label-free PHI (PHotothermal Imaging) and it has been demonstrated in applications to biological samples, with plenty of examples reported in the literature (e.g., Kobayashi, T. et al. *Label-free imaging of melanoma with confocal photothermal microscopy: differentiation between malignant and benign tissue*, *Bioeng.* 5, 67, 2018 – He. J. et al., *Noninvasive, label-free, three-dimensional imaging of melanoma with confocal photothermal microscopy: differentiate malignant melanoma from benign tumor tissue*, *Sci. Rep.* 6:30209, 2016 - Miyazaki J. et al. *Fast 3D visualization of endogenous brain signals with high-sensitivity laser scanning photothermal microscopy*, *Biomed. Opt. Expr.*, 7:5, 2016 – Lasne D. et al. *Label-free optical imaging of mitochondria in live cells*, *Opt. Expr.* 15:21, 2007).

2. *The authors cite scanning thermal microscopy and fluorescence imaging as techniques for high resolution thermometry, yet they leave off scanning thermoreflectance microscopy. While scanning thermoreflectance microscopy is not a sub-diffraction technique, its use of a visible probing laser enables it to readily obtain spatial resolutions ~ 1 μm. It should therefore see an acknowledgement in the literature review of this paper with key citations.*

ANSWER: We thank the Reviewer for pointing out this alternative technical approach, and we apologize for not properly acknowledging it in the original submission. A short description of the technique and related bibliographic references have been added to the manuscript introduction (Rosencwaig, A. et al. *Detection of thermal waves through optical reflectance*, *Appl. Phys. Lett.* 46, 1013, 1985 - Farzaneh M. et al. *CCD-based thermoreflectance microscopy: principles and applications*, *J. Phys. D: Appl. Phys.* 42:143001, 2009 – Pottier, L. *Micrometer scale visualization of thermal waves by photoreflectance microscopy*, *Appl. Phys. Lett.* 64, 1618, 1994 – Kim, D. U. et al. *Quantitative temperature measurement of multi-layered semiconductor devices using spectroscopic thermoreflectance microscopy*, *Opt. Expr.* 13906, 2016).

3. *Can the authors comment on how they considered the view factor in their determination of the surface temperature? It is unclear how the Stefan-Boltzmann law can be used to directly interpret surface temperature when then it is important to determine how many photons are incident on the camera. Simply saying in the Methods that they included the camera-sample distance in the camera software, may mislead readers into thinking they did not consider the amount of radiation emitted by the sample and received by the camera.*

ANSWER: Based on the Reviewer’s comment, a newly added Supplementary Note (Supplementary Note 4 in the revised submission) now clarifies the temperature measurement procedure. We also summarize it here for convenience sake.

When observing an opaque grey-body with emissivity ε at temperature T_{obj} , the thermal camera senses a total radiation power

$$\Phi_{tot} = \varepsilon\tau\Phi_{obj}(T_{obj}) + (1 - \varepsilon)\tau\Phi_{amb}(T_{amb}) + (1 - \tau)\Phi_{atm}(T_{atm}) \quad (2)$$

where:

- $\varepsilon\tau\Phi_{obj}(T_{obj})$ is the radiant power contribution of the grey-body, with $\Phi_{obj}(T_{obj})$ corresponding to the radiant power detected by the thermal camera in the presence of a black-body at the object temperature T_{obj} . τ is the atmosphere transmittance and describes the attenuation of the emitted radiation in the atmosphere across the camera-sample distance d .

- $(1 - \varepsilon)\tau\Phi_{amb}(T_{amb})$ accounts for the atmosphere-attenuated thermal radiation emitted by the surroundings at temperature T_{amb} and reflected by the grey-body with reflectance $(1 - \varepsilon)$. According to Kirchhoff’s law, an emissivity of 1 is assumed for the surroundings (*User’s manual FLIR E40 Series*, FLIR Systems Inc., p. 33-34).

- $(1 - \tau)\Phi_{atm}(T_{atm})$ is the radiant power contribution of the atmosphere at temperature T_{atm} , with emissivity $(1 - \tau)$.

As detailed in Vollmer M. and Moellmann K.P, *Infrared Thermal Imaging. Fundamentals, Research and Applications*, Wiley-Vch, 2010 - p.97-101, based on eq. (2) the radiant power $\Phi_{obj}(T_{obj})$ can be retrieved starting from the measured sensor signal Φ_{tot} as

$$\Phi_{obj}(T_{obj}) = \frac{1}{\varepsilon\tau} \Phi_{tot} - \frac{1-\varepsilon}{\varepsilon} \Phi_{amb}(T_{amb}) - \frac{1-\tau}{\varepsilon\tau} \Phi_{atm}(T_{atm}) \quad (3)$$

In order for $\Phi_{obj}(T_{obj})$ to be evaluated via eq. (3), the grey-body emissivity has to be provided to the thermo-camera acquisition software. The atmospheric transmittance τ is derived instead by the same software once the atmospheric temperature T_{atm} , the relative humidity and the camera-sample distance d are provided as input parameters. Together with T_{atm} , the ambient temperature T_{amb} has to be provided to the camera software to enable computation of the two contributions $\Phi_{amb}(T_{amb})$ and $\Phi_{atm}(T_{atm})$ based on the calibration curve stored in the thermal-camera firmware. Such a calibration provides the relation between detected camera signal and black-body temperature, and is also at the basis of the final conversion of the experimental value $\Phi_{obj}(T_{obj})$ into the object temperature T_{obj} .

It is worth remarking that the camera calibration (provided with the camera by the manufacturer) relates the black-body temperature to the actual *detected* sensor signal. Therefore, all the camera properties (e.g., spectral response of the detector, transmittance of the optics, solid angle of signal collection) affecting the sensor signal are automatically accounted for in the calibration curve. All FLIR imaging systems adopt eq. (2) as general measurement formula (*User's manual FLIR E40 Series*, FLIR Systems Inc., p. 33-34), and the radiant chain assumed in eqs. (2)-(3) has always been a fairly accurate description of our experimental situation.

Therefore, in case the Reviewer is referring to the view factor as the fraction of thermal radiation that gets effectively collected within the solid angle subtended by the camera, no a posteriori correction for the view factor was necessary in our data-sets.

We finally remark the thermography literature formally defines the view factor as the fraction of radiant power emitted by a surface A_1 and intersected by a second surface A_2 , depending on the distance and relative orientation of the two areas with respect to their connecting line. The concept of view factor is therefore typically used when quantifying the net energy transfer that occurs between pairs of objects at different temperatures (Vollmer M. and Moellmann K.P, *Infrared Thermal Imaging. Fundamentals, Research and Applications*, Wiley-Vch, 2010 - p.19-21, 373-375). In the presence for example of an object at temperature T_{obj} which is adjacent to a surface at temperature $T_{amb,1} \neq T_{obj}$, a net energy transfer is established between the object and the surface according to the temperature difference $T_{obj}^4 - T_{amb,1}^4$ and according to the respective view factor. The same reasoning applies to the energy transfer between the object and *all* the j -th surrounding surfaces at different temperatures $T_{amb,j} \neq T_{obj}$, so that, overall, different portions or sides of the object may have different temperatures (more radiation being emitted than received in some parts of the object, and more radiation being received than emitted in other parts). The view factor and the effect of nearby structures are particularly relevant in outdoor building thermography, where neighboring buildings and the sky have a significant influence on the measured surface temperature of a wall or roof and may lead to higher temperature areas being misinterpreted as thermal leakages. By contrast, we usually deal with objects in thermal equilibrium at room temperature (when the sample is not illuminated by the laser beam); even during the laser illumination events, the sample temperature is only at most a few degrees above room temperature. As a consequence, the temperature increases detected in our imaging experiments should not be attributed to the presence of higher-temperature surfaces surrounding the imaged object. Comparison of the super-resolved thermal images we have obtained with the transmitted-light images of the same samples further confirms temperature variations are only induced on light-absorbing and heat-releasing portions of the specimen (for example, ink printed areas of the microfiche patterns), as expected.

4. What is the viewing angle with respect to the sample? Can the authors comment on whether consideration of this will affect their results? This citation describes how the viewing angle has dramatic effects on the measured temperature signal: Ball et al. "Factors affecting the accuracy of thermal imaging cameras in volcanology", *Journal of Geophysical Research*, Vol. 111, B11203, 2006.

ANSWER: We thank the reviewer for raising this important point. All the experiments have been performed with the thermal camera facing the laser-illuminated surface of the sample along the normal direction in the

xz-plane, with the only tilt along the vertical direction never exceeding 20° (i.e., with the thermal camera pointing to the sample along the green arrow in the sketch below, where the adopted imaging geometry corresponds to $\phi=0^\circ$ and $\theta=10^\circ$ - 20°). Such a thermal camera tilt and the resulting viewing angle with respect to the sample might affect (i) the size of imaged objects, distances being subject to the foreshortening effect of perspective view, and (ii) measured temperature values due to the possible angular dependence of the sample emissivity.

(i) Effect on the size of imaged objects. Following the Reviewer's suggestion, the effect of perspective view on imaged object sizes is now quantified and commented in detail in a newly added Supplementary Fig.15. Reported experimental results demonstrate that, even under the imaging configuration corresponding to the highest deviation from normal observation ($\phi=0^\circ$, $\theta=20^\circ$), the 20° thermal camera tilt along the vertical axis enables accurate imaging of object sizes even across large (cm^2) areas of the field of view. Indeed, the thermographic image of a heated object of known size and shape (here, a glass slide $7.65\text{cm} \times 2.5\text{cm}$ in size) only reveals a 3% variation of the object width across the 2.5cm distance along the vertical direction. When the object width on the thermal image (varying from 197 to 191 pixels) is exploited to derive the horizontal thermo-camera pixel size δx_T on the sample plane, this 3% variation only translates into a 2% uncertainty on the pixel size ($\delta x_T=395\pm 8 \mu\text{m}$). As expected instead from the absence of any thermo-camera tilt in the xz-plane, nearly no perspective effect is observed for the imaged height of the glass slide: the 2.5cm imaged object height shows a 1.6% variation across the whole 7.65-cm length, resulting in a 1% uncertainty on the vertical pixel size ($\delta y_T=404\pm 4 \mu\text{m}$). Significantly, the absence of relevant perspective effect over $\sim 20 \text{ cm}^2$ areas definitely excludes foreshortening effects over the smaller $\sim \text{mm}^2$ areas we have imaged in the experiments of Figures 2 and 3. It is finally to be noted that the compatibility between the retrieved δx_T and δy_T pixel sizes along the x- and y- directions justifies our assumption of a square thermo-camera pixel size on the sample plane, to be exploited for the localization of laser-induced temperature variations and image reconstruction during super-resolution imaging experiments.

(ii) Effect on the measured temperature. While black-bodies behave like perfect isotropically diffuse emitters, real grey-body surfaces may display a dependence of the emissivity ϵ on the angle of observation with respect to the surface normal (Ball M. et al. *Factors affecting the accuracy of thermal imaging cameras in volcanology*, J. Geophys. Res. 111, B11203, 2006 – the paper is now cited in the revised manuscript and Supplementary Material) We have therefore quantified the emissivity for the biological specimens with the thermal camera pointing to the sample under the aforementioned 20° vertical tilt (the results are discussed in the answer to Reviewer's comment #11). Furthermore, we have carried out a characterization of the emissivity angular dependence for the ink samples employed for Fig.2 in the broad range $\theta=0^\circ$ - 40° (at fixed $\phi=0^\circ$), following the procedure suggested in the literature (Vollmer M. and Moellmann K.P, *Infrared Thermal Imaging. Fundamentals, Research and Applications*, Wiley-Vch, 2010 - p.38). The description of the measurement procedure and experimental results are reported in Supplementary Note 4 and in the newly added Supplementary Figure 14, demonstrating that the emissivity of our microfiche ink samples ($\epsilon=0.80\pm 0.03$) does not exhibit any appreciable dependence on the angle of observation up to our 20° maximum tilt of the thermal camera. Results agree with the literature, reporting constant emissivity from the normal direction ($\theta=0^\circ$) up to at least $\theta=40^\circ$ - 45° for the majority of materials (Vollmer M. and Moellmann K.P, *Infrared Thermal Imaging. Fundamentals, Research and Applications*, Wiley-Vch, 2010 - p.38). We confirm therefore the thermal camera viewing angle and small tilt were not biasing the reported temperature values.

5. The sentence in line 127 – 129 is not clearly written.

ANSWER: The sentence has now been rephrased to better describe the adopted modulated laser-illumination scheme.

6. All figure captions are not easy to comprehend with too many subfigure numbers. They could be clearer with elaborations.

ANSWER: In the attempt of making them clearer to the reader, we have modified the captions of both Figures 2 and 3 (Figure 2 has been substantially simplified according to Reviewer's comment #10).

7. Can the authors comment on how they optimized their laser spot size? They discuss in detail how smaller spot sizes can improve the imaging resolution, but do not comment on the potential drawbacks of reducing the laser spot size. Sample damage? Low signal?

ANSWER: Indeed, in the original submission we focused on the beneficial effect of a reduced laser spot size on the imaging resolution. There could be some drawbacks in reducing the spot size too much, particularly for biological samples. At fixed laser output power, a higher beam focalization leads to higher intensity and increased risk of sample damage. In the experiments performed here, the intensity on the sample was about 0.64 kW/cm^2 , well below the damage threshold for skin at a radiation wavelength of 630 nm ($\cong 1.4 \frac{\text{kW}}{\text{cm}^2}$, as derived from the EU directive number 2006/25/CE, published on 27 April 2006). Furthermore, the damage of biological specimens was excluded by sequential transmitted-light imaging of the explanted skin biopsies before and right after thermal imaging experiments: results were already cited in the main text and included in the original manuscript as Supplementary Figure 9 (now Supplementary Fig.12). Regarding microfiche samples, no evident damage of the microfiche printed patterns was produced by the adopted laser intensities. However, since the intensity increases quadratically with the beam radius, particular care should be devoted to the evaluation of the damage threshold in case of measurements on a new, different sample.

Regarding the effect of the reduction of the beam spot size on the signal, we do not expect a marked decrease of the thermographic signal for a uniform sample. In fact, the thermal signal is proportional to the number N of absorbers hit by the laser, the absorption cross-section σ of the absorbers and the laser intensity I : $\Delta T \sim N I \sigma$. The intensity is determined by the ratio of the power, P , over the spot area, A : $I = P/A$. By reducing the beam size we are increasing the intensity, but also decreasing the number of micro-absorbers hit by the laser. Overall, if the sample is perfectly uniform, with a surface density of absorbers μ , the thermal radiation (proportional to the temperature difference), would be completely insensitive to the change in the laser spot size: $\Delta T \sim N I \sigma = N \frac{P}{A} \sigma = \mu P \sigma$. There are obvious limitations to this general argument: the surface density might not be perfectly uniform (with an average correlation length L_c), or the sample could be damaged at high intensity values and change its absorption or emission properties. In particular, if the absorbers density changes on the spatial scale of the laser spot size (or $L_c \cong \omega_0$), its reduction will imply a reduction of the thermal emission in proportion to the square of the beam size, or even steeper. This being said, we have not investigated experimentally in detail the effect of the reduction of the beam spot on the emitted power and on the signal/noise ratio of the collected thermograms, based on the observation that for both Figures 2 and 3 no evident effect of the spot reduction was observed neither in terms of photo-damage nor in terms of signal decrease. We recognize however that a study of the temperature increase as a function of the beam size, at constant intensity, could provide us with important information about the surface inhomogeneity (i.e. the correlation length L_c).

8. In line 198 and other places, the authors specify the thickness of the ink stripe as $30 \mu\text{m}$. However, it should be the width, not thickness. This should be corrected. With this pointed out, what is the thickness of the ink stripe? This should be thick enough to exclude the substrate emission effect.

ANSWER: We acknowledge that our use of the term "thickness" is not correct and we have replaced it by the term "width". The thickness of the ink stripe, as estimated by confocal reflectance imaging of a cross-section of the microfiche, is few micrometers (of the order of 1/20 of the microfiche support that is $100 \mu\text{m}$ in overall thickness). The Reviewer's concern regards the possibility that the ink stripe absorbs the visible laser, it is transparent to the region $9\text{-}13 \mu\text{m}$ and (as suggested in the major comment #3) has high thermal diffusivity.

We are not able to evaluate the transmittance of thermal radiation by such a thin layer, but we expect that some of the thermal radiation emitted by the substrate will pass through the thin layer of ink and superimpose to the radiation emitted directly by the heated ink on top. However, as discussed above, even in such situation, our pulsed and raster scanning excitation mode still provides us with the localization of the absorbers within the laser spot (experimental data demonstrate it with Fig.2 and Supplementary Fig.4 of the original submission).

9. The authors should specify the criteria for the approximation in line 207.

ANSWER: Based on Stefan-Boltzmann's law, the number of collected infrared photons scales as $N \propto \varepsilon\sigma(T_{max}^4 - T_0^4)$, where ε is the sample emissivity, σ is Stefan-Boltzmann's constant, T_0 is the sample temperature in the absence of laser illumination and T_{max} is the peak temperature during a laser-illumination event. By introducing the temperature variation $\Delta T_{max} = T_{max} - T_0$ we can write

$$\begin{aligned} N \propto \varepsilon\sigma(T_{max}^4 - T_0^4) &= \varepsilon\sigma((T_0 + \Delta T_{max})^4 - T_0^4) \\ &= \varepsilon\sigma(\Delta T_{max}^4 + 4\Delta T_{max}^3 T_0 + 4T_0^3 \Delta T_{max} + 6\Delta T_{max}^2 T_0^2) \end{aligned} \quad (4)$$

With our typical $T_0=20^\circ\text{C}$ (293 K) and $\Delta T_{max}=2^\circ\text{C}$ (2 K), approximating $(T_{max}^4 - T_0^4)$ to $4T_0^3 \Delta T_{max}$ only results in a 1% underestimate of the true value. More in general, the approximation of eq. (4) holds up to a temperature increase ΔT_{max} of 10 K within a 5% underestimate of the true value for $(T_{max}^4 - T_0^4)$. The underestimate raises to 9.6% with $\Delta T_{max}=20$ K. We have now specified the magnitude of the error associated with the approximation in the main text.

10. Questions about Figure 2:

- How do we trust the temperature scale on the secondary y-axes and image color scales? To strengthen the author's claim that they are quantitatively measuring temperature, they should provide at least finite element calculations to elucidate the theoretical temperature rise expected under the laser illumination conditions. This will verify their assumption on the emissivity of the material and their usage of the Stefan-Boltzmann law.

ANSWER: We agree with the reviewer that knowledge of the sample emissivity is crucial to ensure quantitative measurement of temperature variations, and we apologize for not including the emissivity measurement in the initial version of our manuscript. As anticipated in the answer to point #4, the emissivity of microfiche samples has now been measured with the black-tape procedure (Vollmer M. and Moellmann K.P, *Infrared Thermal Imaging. Fundamentals, Research and Applications*, Wiley-Vch, 2010 - p.46) as described and reported in the newly added Supplementary Figure 14. Then the measured emissivity has been exploited for the analysis of all the datasets of the modified version of Figure 2; all the Supplementary Figures containing microfiche data have been corrected as well.

Provided the reflected emission from ambient sources and the atmosphere contribution are properly taken into account (as described in the answer to point #3), direct measurement of the sample emissivity justifies the usage of the Stefan-Boltzmann law. Implementation of finite element simulations provides a less straightforward method to validate the temperature axis and color scales of the datasets in Fig.2. Accurate simulation of the physics of the system should include implementation of (i) laser light illumination and absorption by the sample, (ii) thermal conduction in the sample itself, (iii) possible convective heat flux towards the embedding medium (air), (iv) emission of thermal radiation by the sample with emissivity ε . Aside from the heat transfer coefficient, this requires knowledge of the sample heat capacity, density, thermal conductivity, emissivity and absorption cross-section, making reliable simulations way harder to achieve than a direct experimental evaluation of the ε parameter.

- The line traces in (d)-(f) and (k)-(l) compare the transmission optical images with the obtained temperature maps. Why are the low temperature data points (i.e., off of the ink features) all at exactly zero? The authors provide y-scale uncertainties for each data point; however, the average value of these data points do not vary across the line traces within these uncertainties? Moreover, why is there no x,y-scale uncertainty shown for the low temperature data points?

ANSWER: The temperature increment observed outside the ink patterns is (up to ~5-20 mW laser power) below the threshold level $\Delta T_{min} = 0.3-0.7$ °C chosen here. This is due to the very low laser light absorption by the microfiche support: it is indeed a difference in the absorption properties of the printed and non-printed regions of the sample that is at the basis of the discrimination of the ink structure over the polyester substrate. The threshold is applied to the reconstructed image only to remove very low and spurious temperature variations that might be erroneously detected by the peak identification algorithm even in the absence of light absorption on noisy frames at constant temperature. The threshold does not eliminate any true laser-induced temperature variation, since the typical ΔT_{max} values are at least twice as high as ΔT_{min} (notice this applies to both the data-sets of the new version of Figure 2).

The y-scale uncertainty we have adopted for measured temperature values coincides with the 0.1°C thermal camera sensitivity as stated in the text. Data points vary within these uncertainties across temperature profiles in agreement with the decreasing absorption of laser light at the ink borders revealed by transmitted-light images. The x-scale uncertainty has been derived from Supplementary Fig.4e (now Supplementary Fig.5e) and is a function of the detected temperature variation. As a result, no uncertainty has been reported in the sample areas where no temperature increase has been experimentally detected. All these observations apply to both the original and new versions of Fig.2.

- *The authors select seemingly random lines from the temperature images for the line traces in (d)-(f) and (k)-(l). It appears that only temperature line traces that match the best with the optical line traces are selected. While the temperature image is somewhat able to discern the “LABS” text, it is not as highly resolved as the optical image. However, the selected line traces may mislead in that the temperature resolution is comparable to the optical image resolution.*

ANSWER: Temperature and intensity line traces in panels (d)-(f), (k)-(l) have been initially reported to provide a quantitative comparison of the super-resolved thermal images with the corresponding conventional transmitted-light images of the samples. The only visual inspection of the images would limit to a qualitative evaluation of the reconstruction of the patterns shape and sizes. Of course, our reconstruction is accurate within the resolution limit of the proposed technique (~60-100 μm), and temperature/intensity profiles should not suggest super-resolved thermal imaging could achieve the ~200 nm resolution of transmitted-light microscopy. In order not to mislead the readers, the 200-nm resolution of optical microscopy at 633-nm is now explicitly reported in the Ms. to make this point clear. At the same time, we have simplified Fig.2 and reduced the total number of reported temperature/intensity profiles. We have only reported (i) the profile across the letters “B” and “S” of the “LABS” pattern and (ii) the average profile across the whole stripes grating to quantify the achieved 100 μm and 60 μm resolution in the two cases.

- *The authors claim that the spatial resolution of their thermal image is ~60 μm or smaller. If this is true, Fig. (i)-(i) should clearly distinguish the line patterns separated 60 μm apart. However, they do not clearly distinguish line patterns. The reviewer is not convinced of the authors’ claimed spatial resolution. More careful experiments should be conducted to rigorously define the best spatial resolution of their method, like imaging line patterns with different separations until the obtained thermal images cannot clearly distinguish the line patterns.*

ANSWER: The dataset reported in panels (g)-(l) of the initial version of Fig.2 was acquired with low-power laser illumination ($P=4.5$ mW), resulting in temperature variations in the 0.3-1.4°C range and peak-localization uncertainties as high as 90 μm . Yet under this condition, as revealed by the average temperature profile in panel (l), the signal dip between adjacent temperature peaks already satisfied the 26.4% contrast threshold required by the well-known Rayleigh criterion, which is employed for the definition of a resolution limit in optical microscopy (Stelzer, E. H. K. *Contrast, resolution, pixelation, dynamic range and signal-to-noise ratio: fundamental limits to resolution in fluorescence light microscopy*, J. of Microsc. 189,15-24, 1998). This is why the dataset was originally included in Fig.2. However, as the reviewer correctly remarks, the obtained super-resolution thermal image (panel i in the original version of Fig.2) does not enable a clear visual identification of the pattern lines. We have therefore imaged the same microfiche sample over a larger area (2.45x0.64 mm²) with higher laser power

(18 mW) and reduced scan pixel size (15.3 μm). The dataset, which is reported in the new version of Fig.2, allows clearly distinguishing the lines of the ink pattern. To further strengthen the claimed achievement of 60 μm resolution the reconstructed thermal image is shown together with the vertical average temperature profile, and satisfaction of the Rayleigh criterion is highlighted by explicitly reporting contrast percentages between adjacent temperature peaks (see the revised version of Fig.2). As stated in the manuscript, the achieved resolution is a function of both the laser spot size and the peak localization uncertainty. This is assigned in turn by the number of collected infrared photons, i.e., by the laser power, the sample emissivity and the sample absorption cross-section. Accordingly, results reported in Fig.2 do not aim at quantifying the best spatial (super) resolution of the method, which is not assigned in absolute terms: while confirming a proof-of-principle 20-time gain with respect to the conventional resolution of the thermal camera on microfiche samples, Fig.2 aims at demonstrating that the spatial resolution can be pushed and regulated by acting on the experimental setup and image acquisition parameters in a balance of resolution, sample photo-damage and total data sampling time.

- *Using two color scale bars in one figure is confusing. The authors should select one.*

ANSWER: We thank the reviewer for evidencing the mistake. The same lookup table has now been employed for all the panels of the figure.

- *Using different scale bars without texts in figures is confusing.*

ANSWER: Equal 500- μm length has now been adopted for all the scale bars appearing in the figure.

11. Questions about Figure 3:

- *Is it necessary to show boxes 2 and 3 in the images of Fig. 3(a) and (f), or remove boxes 2 and 3 from (a)? The results of these boxes are not shown in the main text but may confuse a reader who may be looking for those results.*

ANSWER: Following the Reviewer's suggestion, we have removed ROIs 2 and 3 in Fig.3a. However, it is only by showing the thermal images of all the ROIs (1, 2 and 3) that – as claimed in the text – we can provide a complete characterization of the distribution of PBNPs inside the tissue via photo-thermal imaging. Without reporting the thermal images of ROIs 2 and 3, the presence of PBNPs outside ROI 1 could not be excluded based on thermal data alone, requiring instead inspection of the transmitted-light image of the same treated skin biopsy. We have therefore preserved the analyses of the datasets acquired on ROIs 2 and 3, which are now only mentioned in the main text (not in the figure) and included as Supplementary Figure 10.

- *In particular, why have the authors not included the results of boxes 2 and 3 from Fig. 3(f)? They include the results of box 1 into Fig. 3(g), but this section has the least biological sample. Can the authors explain why this choice was made?*

ANSWER: The analysis of ROI 1 in panel (f) was originally included in Fig.3 for symmetry with respect to the treated sample: PBNPs localize above and at the border of the treated skin section (Fig.3a), in a position that would roughly coincide in the untreated sample with the area covered by ROI 1. The choice was therefore intended to inspect the presence of temperature increases in analogous regions of treated and control specimens. However, following the Reviewer's comment, we have now included the results of the photo-thermal acquisitions on ROIs 2 and 3 in a newly added Supplementary Figure 11. Furthermore, we exploit the results obtained on all the three ROIs of the untreated biopsy for the histogram of temperature variations in panel (i), as suggested by the Reviewer in the following comment.

- *In addition to the previous comment, the authors compare the histograms of (g) and (b) in Fig. (i) to illustrate the role of the PBNP in increasing the temperature rise of the biological sample; however, there*

is almost no biological sample in (g). Is this really a fair comparison or should the result of either box 2 or 3 in (f) be used?

ANSWER: Coherently with the previous Reviewer's comment and for the sake of completeness, we have now reported in panel (i) the histograms of the temperature variations detected in all the three ROIs of the control sample.

- *The authors should provide more physical discussion about the observed temperature distribution in Fig. (b) and (c). Why does the temperature map vary as it does? It is difficult to understand why certain areas are purple while others are light green. The reviewer suspects that this difference might be due to the different distribution of PB-NPs, but there may be effect from the uneven sample thickness and emissivity distribution. No explanations are given from the authors although this seems to be a very important question.*

ANSWER: We thank the reviewer for the useful comment. When fit to a Gaussian trial function, the distribution of the temperature variations detected in the treated skin biopsy and reported in the histogram of Fig.3i shows a standard deviation of 0.27°C. We agree that this variability of the temperature increments might be due to the uneven distribution of PBNPs, but also to a non-uniform emissivity of the sample across the tissue. We can assume that the sample emissivity is everywhere comprised between: (i) the emissivity ϵ_{glass} of the bare glass coverslip, and (ii) the emissivity ϵ_{PBNP} of a solution of PBNPs cast on the glass coverslip at the very same concentration C employed for the NPs injection in the treated tissue section of Fig.3. Indeed, ϵ_{glass} would apply to all the pixels of the reconstructed image located outside the skin section, whereas ϵ_{PBNP} would apply to the pixels containing the nanoparticles at the highest possible concentration C . Since the remaining pixels would contain an intermediate local concentration of nanoparticles, it is reasonable to assume the emissivity in those pixels could not exceed ϵ_{PBNP} . When the nanoparticles distribution and local concentrations are not known a priori (as in the present case), the range $\epsilon_{\text{glass}}-\epsilon_{\text{PBNP}}$ provides the possible emissivity values for every pixel of the reconstructed thermal image. We have therefore characterized both ϵ_{glass} and ϵ_{PBNP} by the black-tape method, and obtained $\epsilon_{\text{glass}}=0.93$ and $\epsilon_{\text{PBNP}}=0.97$ (note ϵ_{PBNP} does not differ significantly from the emissivity of bare skin, values between 0.95 and 0.98 being reported in the literature for humans, mice and several other mammals; Lahiri, B.B. et al, *Medical applications of infrared thermography: a review*, Infrared Phys. Technol. 55, 221-235, 2012 – Mortola, J.P., *Thermographic analysis of body surface temperature of mammals*, Zool. Sci. 30(2), 118-124, 2103 – Polymeropoulos, E.T. et al, *The evolution of endothermy – from patterns to mechanisms*, Frontiers in Physiol., 2018). Based on these results, we have subsequently adopted an average emissivity of 0.95 for the datasets collected on all murine biopsies. At the same time, we have exploited the range $\epsilon_{\text{glass}}-\epsilon_{\text{PBNP}}$ to quantify, at each pixel of the reconstructed images, how much the uncertainty on the emissivity value propagates to the uncertainty on temperature variations. Significantly, when we analyze the very same dataset of Fig.3e with space-independent emissivities = 0.93, 0.95 or 0.97, nearly identical maps of temperature variations are obtained: the three images are reported in Supplementary Fig.9 to demonstrate that the uncertainty on the emissivity value in the range 0.93-0.97 does not hamper the thermal reconstruction of the nanoparticles distribution inside the tissue, and allows to estimate the local temperature increments with a maximum uncertainty of 0.1°C. Based on these considerations, we conclude that the assumption of uniform emissivity does not affect sensibly the temperature values measured in Fig. 3. As now mentioned in the manuscript Discussion, future work will be devoted to extract information on the NPs concentration starting from measured temperature increments and their variability.

- *Clarification question: Are regions that are not attributed a color below a delta temperature of 0.3 K?*

ANSWER: Yes. A clarifying sentence about the ΔT_{min} threshold has now been added to the Methods section of the manuscript.

- *How can (g) be used to evaluate the sample emissivity if there are no PBNP? Can the authors clarify that the PBNP are very transparent in the emission window 7-14 μm ?*

ANSWER: We thank the reviewer for highlighting the mistake from our side. The whole sentence has now been removed from the Methods section, and a reference to Supplementary Fig.14 – reporting emissivity measurements – has been added together with the emissivity values of both synthetic and biological samples.

- *How do the authors accurately determine the emissivity of each of their sample materials? Can they justify why selecting 0.95 is appropriate for each experiment?*

ANSWER: As mentioned previously, the emissivity of microfiche samples and the emissivity of PB nanocubes cast on a glass coverslip have been characterized via the black-tape method. Results are reported in Supplementary Fig.14 for microfiche samples and explicitly mentioned in the manuscript main text and Methods section.

SIMPLE COMMENTS:

1. *Line 161 – error (broken font) in the parentheses*
2. *The paragraphs on p. 10 are unnecessarily short leading to choppy reading.*
3. *In line 203, “lye” seems like a typo of “lie”.*
4. *Lines 222-226 contain redundant statements. Can the authors make this discussion more concise?*

ANSWER: All the typos have been amended. We have also modified the paragraphs on p.10 and shortened the discussion of lines 222-226 as requested by the reviewer.

5. *Line 357: It is unclear what is meant by “reducing the resolution twice”. Technically, making the resolution smaller by two times results in a fourfold increase of the number of pixels and resulting acquisition time.*

ANSWER: We agree with the reviewer the sentence is misleading, and we have rephrased it in the manuscript conclusion.

Reviewers' comments:

Reviewer #1 (Remarks to the Author):

According to the revised version, the manuscript can be accepted after addressing concerns below.

1. I'm still not convinced with advantages of the techniques reported by authors. Actually, commercial thermorefectance imaging equipment can achieve spatial resolution in sub-100 micrometers and high temperature resolution for large area application. In addition, the thermal imaging setup seem to be complicated and have no advance on cost. The authors should clarify these in the manuscript.

2. Another concern is about transient resolution of this thermal imaging technique. In general, we need both steady state and temporal temperature distribution. In the revised manuscript and supplementary material, the experimental response time under laser pulse is several seconds. I think such long time is not suitable for bio-relevance application. I suppose the time constant for temperature detection is related to thermal conductivity (or thermal diffusivity) and optical pulse of pump (probe) beam. How did authors select dwell time when scanning the image? What's the limitation of consuming time for a \sim mm² area imaging? These points should be discussed in the main manuscript.

Reviewer #2 (Remarks to the Author):

The revised manuscript is much clearer and reflects all reviewers' concerns and questions. The authors' response also clearly answers the reviewer's long questions/comments.

The reviewer apologizes for the misunderstanding of the authors' using sub-diffraction in their manuscript. From the authors' clear answers, the reviewer agrees on that their work is a sub-diffraction thermal imaging.

The SI also addresses full details of the substrate effects and the viewing angle effects on the reported scheme, which the reviewer believes are very important.

The reviewer recommends the publication of the revised manuscript for Nat. Comm.

Answers to the comments of Reviewer #1.

REVIEWER'S COMMENT: According to the revised version, the manuscript can be accepted after addressing concerns below. 1. I'm still not convinced with advantages of the techniques reported by authors. Actually, commercial thermorefectance imaging equipment can achieve spatial resolution in sub-100 micrometers and high temperature resolution for large area application. In addition, the thermal imaging setup seem to be complicated and have no advance on cost. The authors should clarify these in the manuscript.

ANSWER: According to the Reviewer's comment we have now better clarified the manuscript Introduction and Discussion sections, and expand here our analysis.

In particular, scanning thermorefectance imaging relies on the measurement of the relative change in the reflectivity of a sample surface as a result of a modulation of the sample temperature. In the basic configuration, while a signal (voltage) generator induces the temperature modulation on the sample, the reflected light of a scanning focused laser or LED probe source is detected in a lock-in scheme by a photodiode or visible-light CCD camera. The measured reflectivity changes are subsequently converted into sample temperature variations by calibration of the proportionality coefficient, which strongly depends on the sample material and composition, illumination wavelength and illumination angle (Farzaneh, M. et al., *CCD-based thermorefectance microscopy: principles and applications*, J. Phys. D: Appl. Phys., 42, 143001, 2009 – Grauby, S. et al., *Comparison of thermorefectance and scanning thermal microscopy for microelectronic device temperature variation imaging: calibration and resolution issues*, Microel. Reliab., 48, 204-211, 2008). Based on these working principles, two important observations can be drawn regarding the technique applications and the cost and complexity of the experimental setup:

(i) Being especially suited to the characterization of high-reflectivity materials (e.g., metals), thermorefectance imaging finds its best application in the characterization of electronic and opto-electronic devices at the micro-scale (Farzaneh, M. et al., *CCD-based thermorefectance microscopy: principles and applications*, J. Phys. D: Appl. Phys., 42, 143001, 2009 – Pierścińska, D., *Thermorefectance spectroscopy – analysis of thermal processes in semiconductor lasers*, J. Phys. D: Appl. Phys., 51, 013001, 2018). To our knowledge, applications of thermorefectance imaging with biological or biotechnological relevance have been reported in the context of the thermal conductivity characterization of solid protein films and cross-linked protein networks, that however needed to be metal coated (Tomko, J.A., et al., *Tunable thermal transport and reversible thermal conductivity switching in topologically networked bio-inspired materials*, Nat. Nanotechnol., 13, 959-964, 2018 – Foley, B.M., et al., *Protein thermal conductivity measured in the solid state reveals anharmonic interactions of vibrations in a fractal structure*, J. Phys. Chem. Lett., 5, 1077-1082, 2014). Therefore, while outperforming infrared thermography for the characterization of low-emissivity metallic devices, thermorefectance microscopy appears less suited to soft biological samples, with the few applications reported in the literature requiring a sample coating with metal (e.g., Al) thin transducer layers to ensure the necessary sensitivity (Yang, J., et al., *Thermal property microscopy with frequency domain thermorefectance*, Rev. Sci. Instrum., 84, 104904, 2013). We remark that, on the contrary, biological application of our super-resolution infrared thermal imaging approach has already been exemplified on explanted murine tissue sections with the dataset of Fig.3. Sub-diffraction thermography only requires the presence of (endogenous or exogenous) light-absorbing and heat-releasing entities in the sample: it is therefore amenable to label-free operation, and offers minimum sample invasiveness for the presented in-vitro and future in-vivo applications to biological specimens.

(ii) A certain variability both in complexity and in cost characterizes experimental thermorefectance microscopy setups. In the simplest configuration, the setup is constituted by the focused scanning LED illumination source, the photodiode detector or CCD camera, and the temperature modulation scheme that is phase-locked to the detector in lock-in configuration. Multi-color excitation, or dispersive spectrometers and white sources, are often required for improved spectral characterization of the thermorefectance calibration coefficient (Kim, D.U., et al., *Quantitative temperature measurement of multi-layered semiconductor devices using spectroscopic thermorefectance microscopy*, Opt. Expr., 24, 13, 2016). The acquired images typically cover $\sim 100 \times 100 \mu\text{m}^2$ fields of view with diffraction-limited micron-sized resolution, with a typical data acquisition time in the 10-minutes range to provide adequate statistics (Pierścińska, D., *Thermorefectance spectroscopy – analysis of thermal processes in semiconductor lasers*, J. Phys. D: Appl. Phys., 51, 013001, 2018 - Grauby, S. et al., *Comparison of thermorefectance and scanning thermal microscopy for microelectronic device temperature variation imaging: calibration and resolution issues*, Microel. Reliab., 48, 204-211, 2008). A dramatic increase in cost and complexity is associated to more sophisticated

implementations of both time-domain and frequency-domain thermoreflectance methods, aimed at extending the imaging capability to the quantification and spatial mapping of thermal conductivities and at the realization of transient, high time resolution, thermal imaging by means of ultrafast pulsed laser sources (e.g., Ti:Sapphire lasers) in pump-and-probe configuration (Yang, J., et al., *Thermal property microscopy with frequency domain thermoreflectance*, Rev. Sci. Instrum., 84, 104904, 2013 – Natesan, H., et al., *Multiscale thermal property measurements for biomedical applications*, ACS Biomater. Sci. Eng., 3, 2669-2691, 2017).

We believe the setup we have employed for sub-diffraction thermography in the present work does not exceed in complexity or cost the simplest commercially available benchtop thermoreflectance microscopes. Indeed, as described in the main text and schematically depicted in Supplementary Fig.3, our components comprise a continuous-wave visible/near-infrared laser beam, a commercially available thermal camera, an electrically controlled shutter for modulated illumination and a beam focusing and scanning system. This results in a total estimated setup cost of ~20k€, which is now specified in the manuscript Discussion. The optical path is compact and easily aligned, and is substantially simpler than the pump-and-probe configurations of both PHI (photothermal imaging) and transient thermoreflectance setups.

REVIEWER'S COMMENT (cont'd): 2. Another concern is about transient resolution of this thermal imaging technique. In general, we need both steady state and temporal temperature distribution. In the revised manuscript and supplementary material, the experimental response time under laser pulse is several seconds. I think such long time is not suitable for bio-relevance application. I suppose the time constant for temperature detection is related to thermal conductivity (or thermal diffusivity) and optical pulse of pump (probe) beam. How did authors select dwell time when scanning the image? What is the limitation of consuming time for a ~mm² area imaging? These points should be discussed in the main manuscript.

ANSWER: The proposed imaging approach achieves super-resolution information by the a-posteriori accurate localization of a series of sparse laser-primed temperature increments. As the Reviewer points out, it is therefore limited in the possibility of providing the simultaneous temperature measurement at all spatial locations across the sample. The same intrinsically applies to all the fluorescence-based super-resolution techniques (e.g., PALM and STORM) relying on a stochastic molecular switching and readout: the data acquisition time was as long as 2-12 hours in the first reported biological application of the PALM technique (Betzig, E., et al., *Imaging intracellular fluorescent proteins at nanometer resolution*, Science, 313, 2006), and it typically remains in the minutes range in the most recent implementations based on deep-learning (Zhuang, X., *Nano-imaging with STORM*, Nat. Photonics, 3, 2009 - Nehme, E., et al., *Deep-STORM: super-resolution single-molecule microscopy by deep learning*, Optica, 5, 4, 2018 – Strack, R., *Deep learning advances super-resolution imaging*, Nat. Meth., 15, 403-409, 2018 – *Advanced optical methods for brain imaging*, ed. Springer). Similarly to the case of PALM/STORM, whose applicability to biological systems is well-established, we reason that a number of bio-relevant applications would benefit from the achievement of super-resolution information while not necessarily requiring transient thermal imaging at high temporal resolution. Applications in the biological and biomedical field for our sub-diffraction thermal imaging approach can be envisioned for example in the characterization of the distribution of metallic nanoparticles in explanted biopsies for the optimization of photo-thermal therapies, or in the ex-vivo development of active thermography pre-clinical protocols in the context of melanoma screening and diagnosis. These applications have been outlined in the manuscript Introduction and Discussion and do not demand for imaging times shorter than those currently allowed by our technique.

In the present realization of our imaging approach, at fixed scan pixel and image size, the total acquisition time is determined by the laser activation time at each illumination event, and the time required to ensure thermal relaxation in between consecutive heating events. As can be deduced from Supplementary Fig.1, for a given induced temperature increment, neither the thermal relaxation time nor the sample characteristic heating time is significantly affected by the sample thermal diffusivity. Therefore, the laser activation time can be selected by means of a simple practical criterion irrespectively of the sample thermal properties: based on the desired temperature peaks amplitude (i.e., the desired signal-to-noise ratio, which will impact in turn on the peaks localization uncertainty and the achievable resolution), we adopt the minimum laser illumination time and the maximum laser power that ensure the achievement of such a desired signal amplitude while avoiding any sample photo-damage. Once the laser activation time has been set, the minimum spatial distance and time interval between pairs of consecutive illumination events are specified according to the sample thermal properties following the criteria we detail in Supplementary Note 1.2.

Overall, we demonstrate with the dataset of Fig.3 that the adopted modulated illumination scheme allows scanning a $3.7 \times 1.8 \text{ mm}^2$ area with 1-s laser illumination time and $75\text{-}\mu\text{m}$ pixel size in 20 minutes. We are currently working on a reduction of the imaging time as a further improvement of the presented technique. The acquisition time can be reduced by acting on the setup hardware with SLM (Spatial Light Modulator)-based multi-spot illumination, and/or by relaxing the constraint on the observation of individual isolated temperature increments. Classes of algorithms allow handling multiple and overlapping signal peaks, based on machine-learning, sequential fitting, sparsity, or maximum-likelihood estimation and could be adopted to speed up the imaging process (Nehme, E., et al., *Deep-STORM: super-resolution single-molecule microscopy by deep learning*, *Optica*, 5, 4, 2018 – Huang, F., et al., *Simultaneous multiple-emitter fitting for single molecule super-resolution imaging*, *Biomed. Opt. Expr.*, 2, 5, 2011).

References to deep-learning and ghost-imaging methods were already included in the manuscript Discussion. Following the Reviewer's comment, this section has been clarified (lines 381-396 in the revised version) to better discuss the temporal resolution of the proposed imaging procedure and the bio-relevance of the method. The explicit expression of the total imaging time, that was initially included in the Supplementary Material only, has now been inserted in the main text (lines 168-169). Similarly, the criteria we have adopted for the selection of the scanning parameters have been specified in the Methods section of the manuscript, as requested.